

# GreenHouse gas Observations of the Stratosphere and Troposphere (GHOST): an airborne shortwave infrared spectrometer for remote sensing of greenhouse gases

Neil Humpage[1], Hartmut Boesch[1,2], Paul I. Palmer[3,4], Andy Vick[5,6],
Phil Parr-Burman[5], Martyn Wells[5], David Pearson[5], Jonathan Strachan[5], and
Naidu Bezawada[5]

[1]Earth Observation Science, Department of Physics and Astronomy, University of Leicester, Leicester, UK
[2]National Centre for Earth Observation, Leicester, UK
[3]School of GeoSciences, University of Edinburgh, Edinburgh, UK
[4]National Centre for Earth Observation, Edinburgh, UK
[5]Science and Technology Facilities Council, UK Astronomy Technology Centre, Edinburgh, UK
[6]Now at Science and Technology Facilities Council, Rutherford Appleton Laboratory, Oxfordshire, UK

*Correspondence to:* Neil Humpage (nh58@le.ac.uk)

**Abstract.** GHOST is a novel, compact shortwave infrared grating spectrometer, designed for remote sensing of tropospheric columns of greenhouse gases (GHGs) from an airborne platform. It observes solar radiation at medium to high spectral resolution (better than 0.3 nm) which has been reflected by the Earth's surface, using similar methods to those used by polar orbiting satellites such as the

JAXA GOSAT mission, NASA's OCO-2, and the Copernicus Sentinel-5 Precursor. By using an original design comprising optical fibre inputs along with a single diffraction grating and detector array, GHOST is able to observe $CO_2$ absorption bands centred around 1.61 μm and 2.06 μm (the same wavelength regions used by OCO-2 and GOSAT) whilst simultaneously measuring $CH_4$ absorption at 1.65 μm (also observed by GOSAT) and $CH_4$ and CO at 2.30 μm (observed by Sentinel-5P).

With emissions expected to become more concentrated towards city sources as the global population residing in urban areas increases, there emerges a clear requirement to bridge the spatial scale gap between small-scale urban emission sources and global scale GHG variations. In addition to the benefits achieved in spatial coverage through being able to remotely sense GHG tropospheric columns from an aircraft, the overlapping spectral ranges and comparable spectral resolutions mean

that GHOST has unique potential for providing validation opportunities for these platforms, particularly over the ocean where ground-based validation measurements are not available. In this paper we provide an overview of the GHOST instrument, calibration and data processing, demonstrating the instrument's performance and suitability for GHG remote sensing. We also report on the first GHG observations made by GHOST during its maiden science flights on board the NASA Global Hawk



unmanned aerial vehicle, which took place over the eastern Pacific Ocean in March 2015 as part of
the CAST/ATTREX joint Global Hawk flight campaign.

## 1  Introduction

The Paris Agreement (2015) describes a framework for mitigating global anthropogenic greenhouse
gas (GHG) emissions. The success of this agreement lies in our ability to monitor progress on
Nationally Determined Contributions (NDCs) that describe country emission targets. This will be
achieved by global stock takes every five years from 2023. The Paris Agreement emphasizes, in
particular, the requirement of independent verification of emissions or monitoring independent data.
One approach to independently assessing GHG emissions is to monitor atmospheric concentrations
of GHGs. Atmospheric GHGs vary over a range of temporal and spatial scales, reflecting variations
due to their sources, sinks, and atmospheric transport. Here, we describe the calibration and perfor-
mance of a new airborne remote-sensing GHG instrument that seeks to fill the spatial measurement
gap between ground-based remote-sensing instruments and satellite data. Data from this airborne
instrument has the potential to link emissions from small-scale (<1 km) sources to the global-scale
atmospheric GHG variations that are currently observed by both ground-based networks and satellite
instruments.

A large number of atmospheric GHG measurements are collected daily across the globe. Each
type of atmospheric GHG measurement has its advantages and disadvantages. Global-scale *in situ*
measurement networks were designed to observe precisely and accurately large-scale changes in
atmospheric GHGs (e.g. NOAA ESRL, https://www.esrl.noaa.gov/). They provide invaluable in-
formation on baseline values, and records are typically collected over many decades. These have
recently been supplemented by smaller, denser networks within individual countries (Henne et al.
2016, Palmer et al. 2017) and in some cases within individual cities whose purpose is to quantify
national, regional, and local GHG emissions (Bréon et al., 2015; Davis et al., 2017). Similar *in situ*
instruments are installed on atmospheric research aircraft (Pitt et al., 2016; O'Shea et al., 2013) and
in some instances commercial aircraft (Brenninkmeijer et al., 2007; Matsueda et al., 2002). These
measurement platforms allow the study of vertical gradients but are typically deployed for short
durations or over limited spatial domains, although there are notable exceptions. Data collected by
commercial aircraft represent valuable knowledge of variations in the upper troposphere along air
corridors but also provide vertical profiles during airport ascents and descents. In addition to these
*in situ* observations, temporary networks of portable upward-looking Fourier Transform spectrom-
eters (Gisi et al., 2012) have been used to constrain GHG emissions on a local scale, by making
integrated column measurements of GHGs from different locations around a city, for example (Hase
et al., 2015; Viatte et al., 2017).



Satellite column observations of $CO_2$ and $CH_4$ with the precision necessary to determine their
surface emissions are now available (Crisp et al., 2017; Kuze et al., 2009; Buchwitz et al., 2015;
Feng et al., 2017). Because of the stringent measurement requirements (precisions of $<1$ ppm for
$CO_2$ and $<5$ ppb for $CH_4$), reflecting the small fractional variations introduced by surface sources
and sinks, systematic errors of a few percent in the column amount can, if not properly character-
ized, compromise the usefulness of the inferred GHG fluxes (Chevallier et al., 2014; Feng et al.,
2016). To help characterize these errors a global inter-calibrated network of upward-looking Fourier
Transform spectrometers has been established to validate the space-borne data (Wunch et al., 2011).
Nevertheless these data provide global coverage, subject to clouds and elevated aerosol loading, but
at exclusively one local time of day with a typical repeat frequency of a few days. The integrated
column measurement can help to overcome these weaknesses: a column measurement at any one
time and location is a superposition of sources and sinks from upwind regions at different times
of day (Palmer et al., 2008). This does however make interpretation of the column non-trivial, and
consequently it would be difficult with current measurement techniques, even if properly integrated
through inter-calibration efforts, to address the objectives of the Paris Agreement given the impor-
tance of city and point emission sources.

Aircraft remote-sensing instruments are particularly well suited for mapping out small-scale gra-
dients in GHGs, enabling repeated observations over large point sources and diffuse sources, and
providing a link between ground-based *in situ* measurements and space-borne observations. The
Airborne Research Interferometer Evaluation System (ARIES) is a Fourier Transform Spectrometer
(FTS) that is regularly flown on the UK FAAM (Facility for Airborne Atmospheric Measurements)
BAe-146 atmospheric research aircraft. Recent work has developed a capability to retrieve ther-
mal infrared (TIR) $CH_4$ columns that have helped to map distributions of $CH_4$ gradients over the
UK (Illingworth et al., 2014; Allen et al., 2014). The German Methane Airborne MAPper (MAMAP)
is a two-channel near/shortwave infrared (NIR/SWIR) grating spectrometer (Gerilowski et al., 2011;
Krings et al., 2011) that was designed to fly on a range of aircraft. MAMAP has illustrated the use-
fulness of airborne technology to map $CH_4$ and $CO_2$ emissions from coal mines, landfill sites, and
a North Sea bubble blow-out plume (Krings et al., 2013; Krautwurst et al., 2017; Gerilowski et al.,
2015). CHARM-F ($CO_2$ and $CH_4$ Remote Monitoring-Flugzeug) is an integrated-path differential-
absorption lidar (Amediek et al., 2017) that serves as the airborne demonstrator for the DLR-CNES
MERLIN (Methane Remote Sensing Lidar Mission) concept. CHARM-F simultaneously measures
$CO_2$ and $CH_4$ columns and has flown on the German HALO (High-Altitude, LOng-range) research
aircraft. The potential for using airborne hyperspectral imagers such as the NASA Next Generation
Airborne Visible Infrared Spectrometer (AVIRIS-NG) to detect and quantify localized anthropogenic
sources of $CH_4$ has also been demonstrated recently (Thompson et al., 2015). A further NASA in-
strument, called CARVE-FTS (Carbon in Arctic Reservoirs Vulnerability Experiment FTS), has
been used to observe GHG fluxes from an aircraft over Alaska (Dupont et al., 2012).



In this paper we describe the GreenHouse gas Observations of the Stratosphere and Troposphere (GHOST) short wave infrared (SWIR) grating spectrometer. The principal advantage of the GHOST design (described in Sect. 2) is that it is able to image four different spectral bands using a single diffraction grating and detector array, thus saving on space and weight compared with an equivalent

instrument using a separate grating and detector for each band. This extent of wavelength coverage at high spectral resolution, and subsequently the number of different gases that can be observed, is unique for an airborne remote sensing instrument. As well as the instrument design we also show the first set of results from GHOST science flights on board the NASA Global Hawk, which took place during the CAST-ATTREX flight campaign, and describe the methods behind their production from

raw GHOST data. Section 3 describes the laboratory measurements used to perform radiometric and spectral calibration on GHOST, and assesses its performance. In Sect. 4 we outline the involvement of GHOST in CAST-ATTREX, which was based at NASA Armstrong in California during early 2015. Finally, we present the first GHOST flight results (along with the optimal estimation method used to retrieve them from the calibrated radiance spectra) in Sect. 5, and summarize the paper in

Sect. 6.

## 2    The GHOST Instrument

The optical design behind GHOST was developed by the STFC (Science and Technology Facilities Council) Astronomy Technology Centre during two design studies, both funded by the UK Centre for Earth Observation Instrumentation (http://ceoi.ac.uk/), that demonstrated reductions in size and

weight of SWIR earth observation spectrometers by using technology originally developed for astronomy instruments. The optics use a combination of blocking filters and multiple grating orders to provide spectra from several high resolution bands from across a wide range of wavelengths. As well as the size and weight advantages, the optical fibre feed to the spectrometer provides flexibility in the mechanical layout of the subsystems on different aircraft, thermo-mechanical isolation, and

the possibility of optically isolated calibration light input to the spectrometer.

The GHOST instrument, initially designed to meet the strict engineering requirements for installation on the NASA Global Hawk unmanned aerial vehicle (UAV), takes advantage of an optical fibre feed system to split the optics into two units; the Target Acquisition Module (TAM) and the Spectrometer Module. Figure 1 shows how these two components were mounted on a pallet on the

underside of the Global Hawk, along with the Air Transport Rack (ATR) which houses the electronics and support equipment for the instrument. In this section we discuss each of these three components in turn.




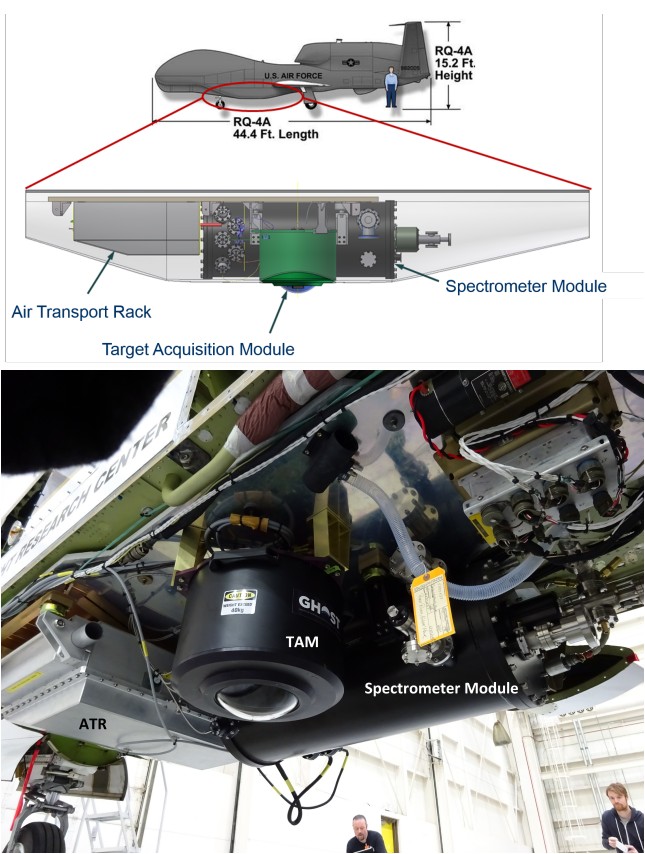

**Figure 1.** Top: Layout of the three main GHOST components as mounted in the lower instrument bay of the NASA Global Hawk; Bottom: Photograph taken from below showing the GHOST installation on the Global Hawk.

### 2.1 The Target Acquisition Module (TAM)

The function of the TAM is to direct sunlight which has passed down through the atmosphere, and

then been reflected back upwards by the Earth's surface, onto an optical fibre bundle which transfers the light into the spectrometer module. To achieve this, the TAM has to be able to acquire light from any angle of rotation and up to 70° away from nadir, depending on the solar zenith angle. A custom off-axis telescope arrangement is used, mounted to a custom Newmark GM-6 gimbal (see Fig. 2). A fold mirror reflects the light path along the elevation axis of the gimbal, through a de-polarising unit,

and onto a second fold mirror that directs the light path along the rotation axis. Mounted centrally along this axis are the pre-fibre optics (a field stop and lenses) and the fibre bundle mounting point. The optics create a pupil image on the end of the fibre bundle such that the light is both concentrated



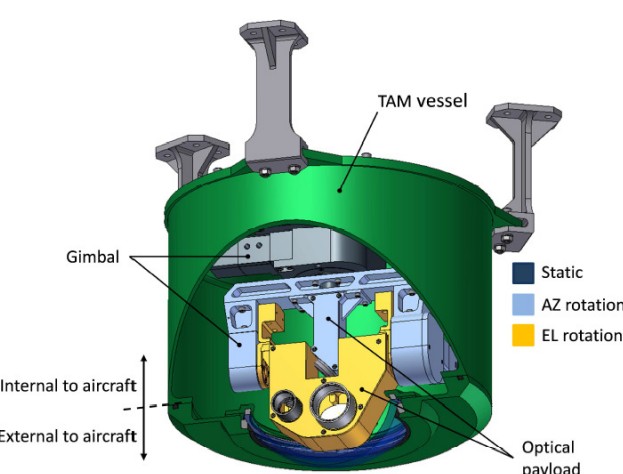

**Figure 2.** Cut-away model of the TAM showing the gimbal, payload, and the edge of the dome (in dark blue). The light blue component rotates about the azimuthal axis, whilst the yellow component rotates about the axis of elevation. The TAM is 513 mm tall including the legs, and has a diameter of 555 mm including the feet.

onto the fibres and evenly distributed between them. The field stop defines the range of angles that will be accepted into the fibres, and is set to an angle of $\pm 6.67°$. By using a design which employs
movable optics to transfer the observed light onto a static fibre bundle, we minimize the potential for deterioration of the measured signal caused by movement and bending of the fibres. The gimbal also carries a wide field pan-chromatic camera with a field of view extending beyond that of the spectrometer input. The camera is used to help interpret the science data, e.g. by identifying when the spectrometer field of view is cloud-free.

The light enters the TAM through a dome whose centre of curvature has been placed at the centre of curvature of the gimbal, to reduce differential aberrations across the gimbal range. Originally the whole bottom face of the TAM vessel was designed to be external to the aircraft, since the vessel was intended as an environmentally benign environment for the gimbal and the optics, with internal heaters and the potential for dry gas flushing. The TAM is pressurized to 3 PSI above local ambient
conditions to minimize the diffusion of water vapour into the optical system. The vessel also included a number of fiducial measurement points which were used to ascertain alignment and location of the optics, both in the lab and on the aircraft. The final installation was more enclosed and left only the dome itself protruding from the underside of the Global Hawk.

     The optical fibre bundle which passes the received light onto the spectrometer module is made
up of 35 multi-mode fibres, each of $0.365\,\mathrm{mm}$ core, encased in a stainless steel outer sheath for protection. At the spectrometer end the bundle of 35 fibres is split into five smaller bundles (one for each spectral band, see Sect. 2.2) comprising seven fibres each, with the choice of fibres kept





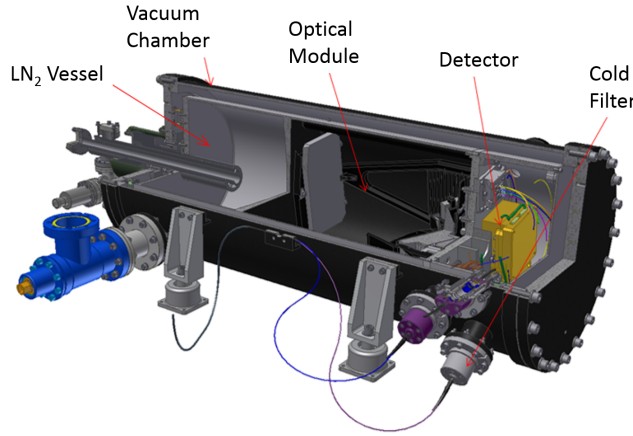

**Figure 3.** GHOST spectrometer module showing the front end compartment, the main spectrometer optics area and the liquid nitrogen cryogenic vessel. The cylinder containing these components is 1064 mm long and 440 mm in diameter. Taking into account the fittings external to the cylinder, the space occupied is 1406 mm long, 758 mm wide, and 440 mm tall.

as spatially random as possible to ensure that a similar sampling of the aperture is passed into each bundle. The light from each of the bundles is transmitted into the (cold) spectrometer through a window and an order sorting filter, which minimises the out of band light, and is finally coupled into a cold fibre bundle.

## 2.2 The Spectrometer Module

Inside the spectrometer module there are three major sections: an end section that contains the optical entry points, cold fibres and the detector box; a middle section containing the spectrometer optics; and a section where the cryogen tank is situated (see Fig. 3). All of the spectrometer module internal components are cooled to $80\,\mathrm{K}$, which has the effect of reducing the thermal background, the dark current on the detector, and any thermo-mechanical instabilities. The decision to use liquid nitrogen for cooling rather than a closed cycle cooler is based primarily on the need to minimise vibrations, both for this instrument and for other instruments mounted on the same aircraft. Cryogenic cooling has the additional benefits of requiring no electrical power, being very reliable (compared with using mechanical cooling which relies on moving components), and using little extra space for external electronics.

The observed light is brought into the spectrometer optics using cold fibre bundles as mentioned in Sect. 2.1. The optical part of the spectrometer module comprises four major components: the inlet slits, fold mirror, camera mirror, and grating (see Fig. 4). The f/5 focused beam from the input slits is reflected by the fold mirror on to one side of the curved camera mirror. The collimated beam is



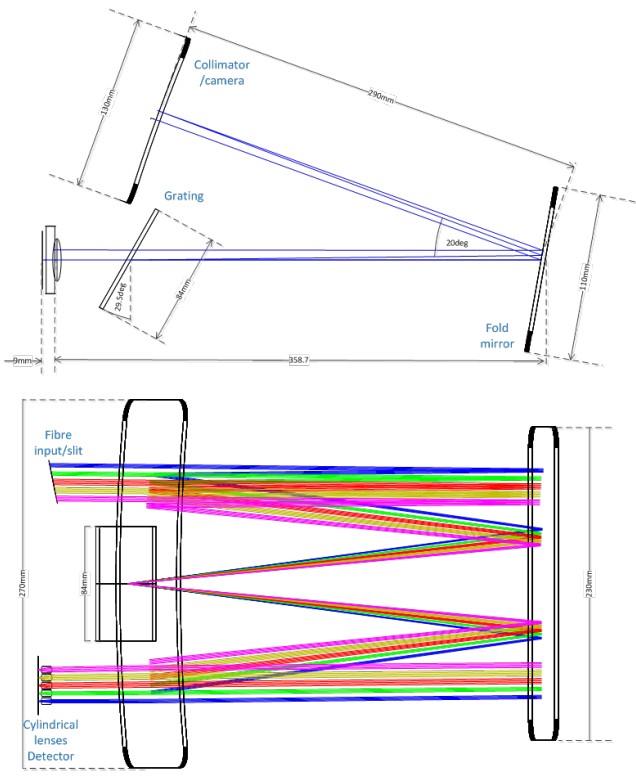

**Figure 4.** Elevation and plan view of the spectrometer showing the major dimensions, which allows it to fit into the cryostat optical module as shown in Fig. 3. The different coloured ray traces in the plan view diagram correspond to light from each of the five optical fibre bundles.

then reflected again by the fold mirror onto the grating, which disperses the light from each band into a number of spectrally resolved images, each produced by a different order of diffraction. The dispersed light is reflected again by the fold mirror on to the other side of the camera mirror, after

which the converging beam is reflected once more by the fold mirror and passed through cylindrical output lenses that concentrate the spectral images onto the detector. The major innovation in this design is the use of multiple grating orders; through choice of orders and positioning of the input slits, multiple high resolution spectral bands can be selected from a wide overall wavelength range and conveniently placed onto the output plane. In the GHOST spectrometer we use a 60 grooves mm$^{-1}$

blazed grating supplied by Richardson Gratings, with a blaze angle of $28.7°$ corresponding to a blaze wavelength of $16\,\mu$m. For spectral bands 1 through to 4 (Table 1) we use diffraction orders 13, 10, 8, and 7 respectively.

    The spectral images are recorded by a Mercury-Cadmium-Telluride (MCT) detector, which is mounted just behind the optics baseplate. The detector used is a Raytheon Vision Systems VIRGO2K





**Table 1.** Target gases, spectral range, spectral resolution and signal to noise ratio (SNR) design requirements for each band. The goal (threshold in brackets) values are given for resolution and SNR. The SNR requirements listed here assume an integration time of 1 second.

| Band | Target gases | Spectral range ($\mu$m) | Resolution (nm) | SNR |
|------|-------------|------------------------|-----------------|-----|
| **1** | $O_2$ | 1.25 – 1.29 | 0.1 (0.1) | 200 (150) |
| **2** | $CH_4$, $CO_2$ | 1.59 – 1.675 | 0.25 (0.3) | 100 (80) |
| **3** | $CO_2$ | 2.04 – 2.095 | 0.15 (0.3) | 150 (100) |
| **4** | CO, $CH_4$, $H_2O$/HDO ratio | 2.305 – 2.385 | 0.25 (0.3) | 100 (80) |

detector with 2048x2048 20 $\mu$m pixels, which has high quantum efficiency (typically greater than 80 % across the broad spectral range), low dark current and low read noise (less than 20 e RMS in a correlated double sample frame). The measured dark current at liquid nitrogen temperature (80 K) is less than 1 e s$^{-1}$ pixel$^{-1}$. During flight we use heaters and a platinum resistor (for feedback) to hold the detector temperature at a slightly warmer 98 K, ensuring that the detector temperature

remains stable to within a few tens of mK via the feedback mechanism (the increased stability is achieved in this way at the expense of a higher dark current, see Sect. 3.2). Prior to installation of the detector array into the spectrometer module, we measured the average signal gain for each pixel to be 4.26 e/count.

The spectrometer has been designed to measure high resolution spectral radiances over four wave-

length bands, which we have chosen to coincide with spectral absorption features of the atmospheric gases we are interested in. The design requirements for the four bands are listed in Table 1, whilst Fig. 5 shows the absorption spectra of the most important gases in each band. When defining the requirements for the spectral resolution and signal to noise ratio we determined threshold values, which indicate the minimum acceptable performance, along with goal values which represent the

desired performance level. In the final design a fifth band, a duplicate of Band 2 (the most important band for meeting our science objectives), was incorporated to provide redundancy in case the original Band 2 did not meet the spectral and radiometric requirements.

### 2.3  Data handling and control

The detector readout system is an ARC (Astronomical Research Cameras) controller attached to an

industrial PC running Linux. The software system is the UK-ATC's 'UCam' product, which has been widely used with astronomy instruments across the world for the last 10 years. The detector operates in an 'integrate while read' mode, a non-destructive readout mode in which pixels can be sampled as many times as required while integrating the signal. At 300 kHz pixel rate, using all 16 amplifier channels, it takes about 0.91 s to read out the entire detector. Since we are only imaging a fraction

of the array, we reduce the readout time by skipping the rows we don't need. By reading out only 300 of the 2048 detector rows (60 rows for each of the five bands), we reduce the readout time to





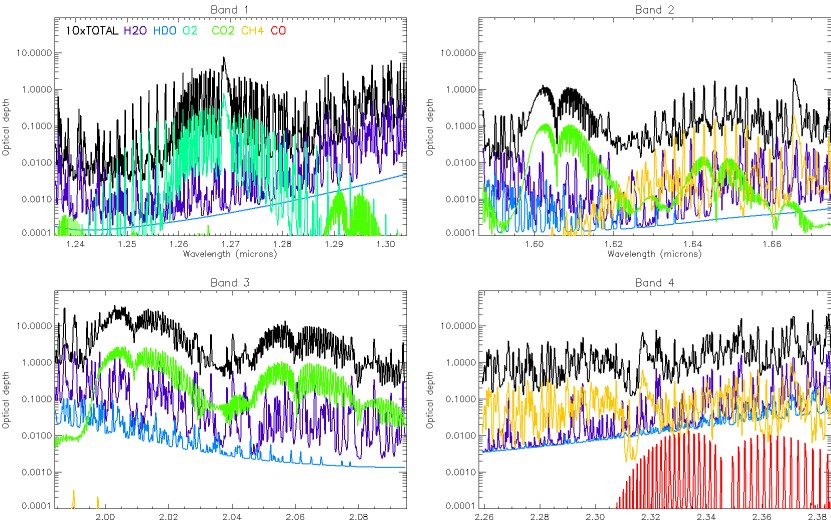

**Figure 5.** Optical depth spectra for each of the four GHOST spectral bands listed in Table 1, calculated assuming a mid-latitude reference atmosphere and a viewing angle of $20°$ off nadir. The different colours correspond to the contributions towards the total optical depth from each species considered. The total optical depth spectrum for each band (shown in black) is multiplied by ten to make it easier to see on the logarithmic scale.

$0.115$ s. In order to maximize the on-source integration time, an up-the-ramp readout mode has been implemented where the array is read out a set number of times following a reset (see Fig. 6). Both the number of read outs between resets and the dwell time between read outs are programmable,

and we set them such that the signal level prior to a reset is below the level at which the detector is saturated. Taking the difference between any two reads results in a frame which is free from the reset noise and the offset.

The electronics for the readout system are contained in the Air Transport Rack (ATR), procured from Elma Electronic in the UK, which provides a temperature and pressure controlled environment.

All data from the system (science data frames, auxiliary pictures, temperatures, positional information, etc.) are stored on a solid state disk in time tagged netCDF files. The system provides TCP (Transmission Control Protocol) and UDP (User Datagram Protocol) command and control access via a 100BASE-T port as well as real time data access (over TCP only). The software system is designed with robustness in mind and a long campaign of simulated operations ensured that the sys-

tem would continue taking meaningful data under almost all conditions and, for extreme eventuality, provided a hard reboot facility.




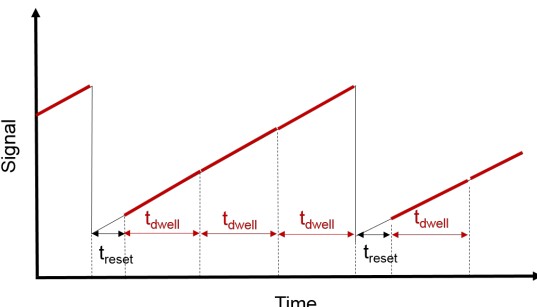

**Figure 6.** The sampling scheme used by the GHOST detector, where we sample the array 'up-the-ramp'; in this example we show three reads of the array subset per exposure. Time $t_{reset} = 0.115\,\mathrm{s}$ is required to perform a destructive readout and reset of the array subset. A non-destructive readout is then performed every $t_{dwell}$ seconds. The number of reads and their duration is determined by the instrument operator.

## 3 Calibration and performance of GHOST

In this section, we describe the procedures and measurements used for the radiometric and spectral calibration of the GHOST instrument, and provide a summary of the spectrometer performance based on the calibration results. We perform the majority of the calibration measurements using a $0.3\,\mathrm{m}$ internal diameter integrating sphere with $0.1\,\mathrm{m}$ diameter output port from Labsphere (Figure 7). The integrating sphere produces a spatially uniform light source, via multiple internal reflections off the inner Spectraflect coating, which completely illuminates the GHOST field-of-view when positioned at a distance of $1\,\mathrm{m}$ from the TAM. The integrating sphere used has four input ports in addition to the output port: three ports housing quartz tungsten halogen (QTH) lamps (two internal to the sphere and one external), and one port allowing the use of alternative light sources (such as the emission lamps used for spectral calibration, as described in Sect. 3.3) with the sphere. The two internal QTH lamps have output powers of 35 and $75\,\mathrm{W}$ respectively, whilst the $150\,\mathrm{W}$ external QTH lamp is mounted behind a mechanical variable aperture which allows fine adjustment of the output luminance. We use measurements of the $150\,\mathrm{W}$ external lamp for the radiometric calibration described in Sect. 3.5. An internally mounted photo-diode provides measurements of the output luminance at a frequency of $1\,\mathrm{Hz}$.

### 3.1 Processing of GHOST measurements into radiance spectra

A number of steps are required to process the raw detector frame data measured by GHOST into radiometrically and spectrally calibrated radiance spectra, which are then suitable for use in retrieval algorithms used to obtain concentrations of target trace gases (as described in Sect. 5.1). We summarize these steps in Fig. 8. The ellipses in the diagram indicate the inputs required, which we obtain



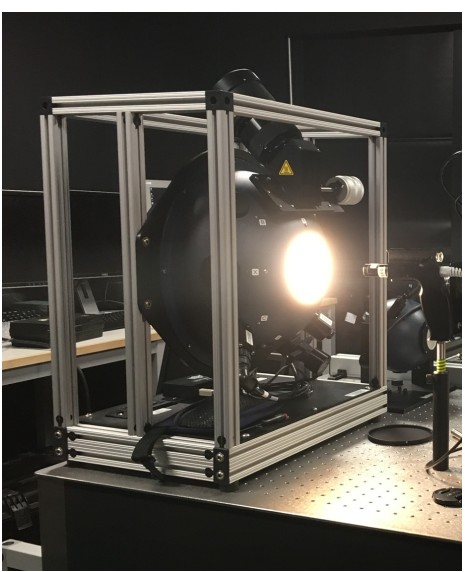

**Figure 7.** Photograph of the NERC FSF integrating sphere, with the 150 W external lamp and variable aperture mechanism visible just above the output port (Photograph by Chris MacLellan, NERC FSF).

during the instrument characterization and calibration described later in this section, whilst the rectangles show each separate processing step in the procedure. The raw detector frames at the start of

the process are obtained by taking the final non-destructive readout in a sequence, and then subtracting the first readout after the previous reset (see Fig. 6). The difference between these two readouts removes the offset and reset noise as described in Sect. 2.3.

## 3.2 Dark current measurement and identification of inactive pixels

Prior to the spectral and radiometric calibration measurements, we need to characterize certain as-

pects of the detector array performance. These include the detector dark current (for each pixel on the array), the identification of the pixels that are illuminated on the detector array by each spectral band image, and the location on the array of dead, hot or otherwise faulty pixels which are excluded from further analysis.

We evaluate the dark current (the residual current which flows through a photosensitive device

in the absence of incident photons) by averaging over exposures taken with no light entering the system, resulting in a dark current map in units of digital counts per second. The mean dark current measured in each band is listed in Table 2. The 'darkness' required for this measurement is achieved by disconnecting the optical fibres from the spectrometer module. The dark exposures are also used to identify hot pixels, defined as pixels which return a fully saturated response independently of

the amount of light incident on them. We flag all pixels with mean measured responses above a



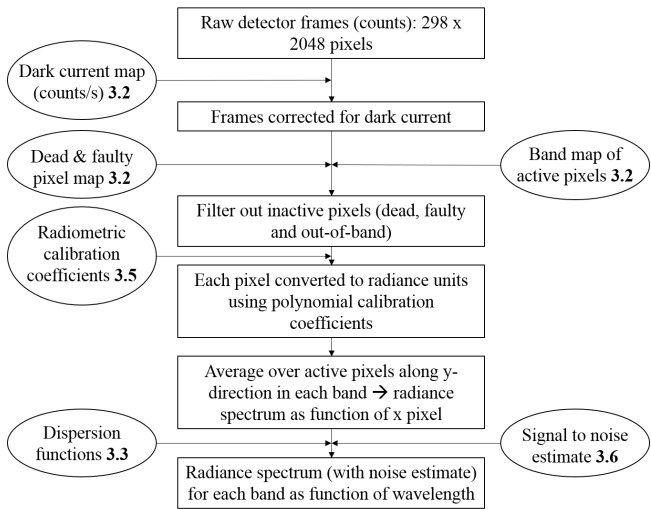

**Figure 8.** Diagram illustrating the steps and inputs involved during the processing of GHOST data into calibrated radiance spectra. The numbers in bold contained in each of the inputs refer to the corresponding section number.

**Table 2.** Mean dark current measured for each GHOST spectral band (see Sect. 3.2) with the detector at flight temperature (98 K). The standard deviations given represent the spatial variability of dark current within each band.

| Band | Mean dark current (counts/s) | Standard deviation (counts/s) |
|------|------------------------------|-------------------------------|
| 1    | 2.855                        | 1.187                         |
| 2A   | 2.375                        | 1.060                         |
| 2B   | 2.698                        | 1.145                         |
| 3    | 2.505                        | 1.021                         |
| 4    | 3.025                        | 1.379                         |

threshold value as hot pixels, and apply a threshold standard deviation to filter out pixels which are 'hot' intermittently, as the pixel response to dark input should not vary significantly over time.

We identify dead pixels, pixels which show no response when light is incident on them, using exposures where a white light source is observed. For this purpose we use the integrating sphere de-
scribed earlier with a quartz tungsten halogen lamp as the observed target. All pixels with measured mean responses below a threshold value are flagged as dead pixels. We also use the same measurements to produce a map of the image of each spectral band on the detector array, that is then used to determine the pixels included when processing the measured images into radiance spectra (see Sect. 3.1). The pixels forming the band map are identified on a column-by-column basis. In each
column, the twenty pixels with the highest value when illuminated (excluding hot and dead pixels)





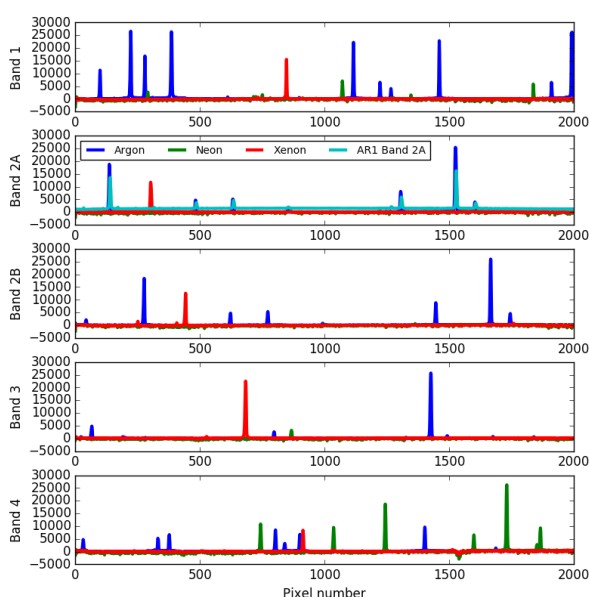

**Figure 9.** Measurements of emission lamp spectra for each GHOST band, shown as digital detector response in counts as a function of horizontal pixel number. The colours correspond to the different emission lamps listed in Table 3.

are considered to be 'in-band' for the purposes of processing the images into spectra. By taking the mean of the twenty in-band pixels in each column during the processing stage (as described in Sect. 3.1) we are able to obtain a better signal to noise ratio for the final calibrated spectra.

### 3.3 Spectral calibration

The aim of the spectral calibration is to determine a relation between the pixel number along the horizontal axis of the detector array, and the wavelength of light incident on that pixel, following dispersion of the input light by the reflection grating. To obtain these relations for each of the five bands we use measurements of emission lamp spectra, where the wavelengths of the emission lines are well known, to evaluate the wavelength of pixels corresponding to the emission line centres.

Figure 9 shows the emission lamp spectra measured by GHOST that we used for the spectral calibration. We use Argon, Neon and Xenon pen-style emission lamps (belonging to the NERC FSF) in conjunction with the integrating sphere to produce spectrally discrete light sources which fill the GHOST field-of-view. The measurements we made for the spectral calibration are listed in Table 3.




**Table 3.** Measurements used for the spectral calibration of GHOST (see text in Sect. 3.3).

| Measurement | Exposures | Reads per exposure | Dwell per read (ms) |
|---|---|---|---|
| Argon pen-style lamp | 100 | 12 | 10000 |
| Neon pen-style lamp | 40 | 12 | 20000 |
| Xenon pen-style lamp | 40 | 12 | 10000 |
| AR1 Argon lamp (Band 2A fibre input only) | 40 | 30 | 1000 |

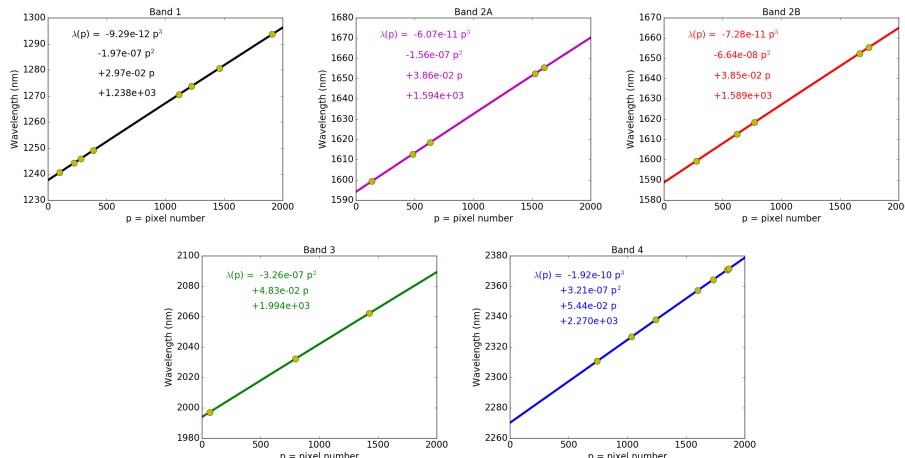

**Figure 10.** Dispersion functions calculated for GHOST Bands 1, 2A, 2B, 3 and 4. Argon emission lines are used for all except Band 4, where neon emission lines are used instead. We used neon for Band 4 because the identification of which measured line corresponds to which line listed in the NIST database was much less ambiguous than for argon in this wavelength range.

We then fit a third order polynomial which returns the wavelength of each emission line as a function of pixel number, giving us a dispersion function for each band. We locate the positions of the line centres in pixel space by first fitting and subtracting any continuum signal in the spectrum, and then fitting an asymmetric Gaussian function to each emission peak greater than a threshold value. We finally obtain the dispersion functions by polynomial fitting of the emission line wavelengths listed in the National Institute of Standards and Technology database (NIST, Kramida et al., 2017) to their corresponding locations in pixel space for each band, as shown in Fig. 10.

### 3.4 Estimate of instrument line shape functions

The instrument line shape (ILS) function is a representation of how a monochromatic light source is observed by the spectrometer, and is an important parameter for calculating the simulated measured spectra used in retrieval algorithms (see Sect. 5.1). In the absence of a genuinely monochromatic





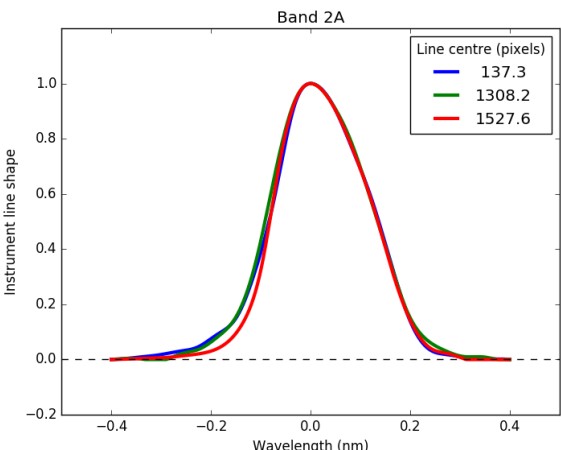

**Figure 11.** Instrument line shapes for band 2A estimated from argon emission lamp measurements. The numbers in the legend give the location in pixel space for each emission line (see second panel of Fig. 9).

source such as a laser, we use measurements of the emission lamps described in Sect. 3.3 with the assumption that the emission lines produced are sufficiently narrow in wavelength that they may be considered to be 'monochromatic' for our purposes.

To obtain the ILS functions for GHOST we first take emission lamp spectra shown in Fig. 9 and apply the spectral and radiometric calibrations described in Sections 3.3 and 3.5 respectively. Once 305    the background continuum signal has been subtracted, three emission lines in each band are selected (such that they are representative of the band wavelength range), extracted, and normalised. The part of the spectrum extracted for each line covers a wavelength range of $\pm 0.4$ nm centred on the maximum. Figure 11 shows the extracted ILS functions for Band 2A. During the forward model calculation described in Sect. 5.1, we interpolate between the extracted instrument line shapes to 310    estimate the ILS for each pixel.

### 3.5   Radiometric calibration

The objective of the radiometric calibration is to determine the relationship between the digital detector array response and the spectral radiance (in SI units) incident on the GHOST fore-optics housed in the TAM, on a pixel by pixel basis. We use the NERC FSF integrating sphere (described previ- 315    ously at the beginning of Sect. 3.3) to produce a spatially uniform light source filling the GHOST field-of-view. The 150 W external QTH lamp is used in conjunction with a mechanical variable aperture to produce high-, mid- and low-level luminances for each band (listed in Table 4). We choose the luminances such that, for each band:





**Table 4.** Measurements of the NERC FSF integrating sphere used for the radiometric calibration of GHOST. The external 150 W lamp was used with a variable aperture to produce the output luminances listed.

| Band | High Luminance | Mid Luminance | Low Luminance |
|------|----------------|---------------|---------------|
| 1 | $7218.14\,\mathrm{cd\,m^{-2}}$ | $3597.69\,\mathrm{cd\,m^{-2}}$ | $719.14\,\mathrm{cd\,m^{-2}}$ |
| 2A | $1007.56\,\mathrm{cd\,m^{-2}}$ | $500.79\,\mathrm{cd\,m^{-2}}$ | $100.48\,\mathrm{cd\,m^{-2}}$ |
| 2B | $1002.77\,\mathrm{cd\,m^{-2}}$ | $500.89\,\mathrm{cd\,m^{-2}}$ | $100.19\,\mathrm{cd\,m^{-2}}$ |
| 3 | $1398.78\,\mathrm{cd\,m^{-2}}$ | $699.92\,\mathrm{cd\,m^{-2}}$ | $140.21\,\mathrm{cd\,m^{-2}}$ |
| 4 | $1796.94\,\mathrm{cd\,m^{-2}}$ | $899.96\,\mathrm{cd\,m^{-2}}$ | $180.15\,\mathrm{cd\,m^{-2}}$ |
| **Number of detector exposures** | 100 | | |
| **Number of reads per exposure** | 10 | | |
| **Dwell time for each read (ms)** | 300 | | |

– The full dynamic range of the in-band pixels is covered between the three luminance levels, and;

– There is some overlap in the dynamic range covered by measuring each luminance level, i.e. the final three to four read-outs in exposures of the low-level luminance output return a similar signal to the first three to four read-outs in exposures of the mid-level luminance output.

For each set of measurements we only connect the optical fibre input for the target band, to avoid saturation of the more optically sensitive bands (Bands 2A and 2B being the most sensitive) when the measured input light reaches the high end of the dynamic range of the less sensitive bands (particularly Band 1). The upper panel of Fig. 12 shows how the measured signal (averaged over 100 exposures) for a single pixel in Band 2A increases with time for each of the three luminances. The signal is read from the detector array ten times per exposure, with each read-out represented by a yellow circle on the plot.

We use a calibration curve supplied with the integrating sphere to estimate the spectral radiance output (in $\mathrm{photons\,s^{-1}m^{-2}\mu m^{-1}sr^{-1}}$) from the luminance measured by the internal photodiode. Assuming that the integrating sphere output is constant with time, we then obtain the time integrated spectral flux for a single pixel as a function of the signal (in counts) measured by that pixel, as shown in the lower panel of Fig. 12. The GHOST calibration curves, shown here for a single pixel in each band, are derived by fitting a fifth order polynomial to thirty data points. These comprise data from ten read-outs (averaged over 100 exposures) of each of the three luminances measured using each band. Calibration coefficients describing the polynomial fit are calculated for every in-band pixel, and saved for use in producing radiometrically calibrated radiance spectra from the GHOST flight measurements as described in Sect. 3.1. The calibration curves also illustrate the signal level at which the detector response begins to show non-linear behaviour; this is at measured signals of around 26000 counts and above for the pixels shown.





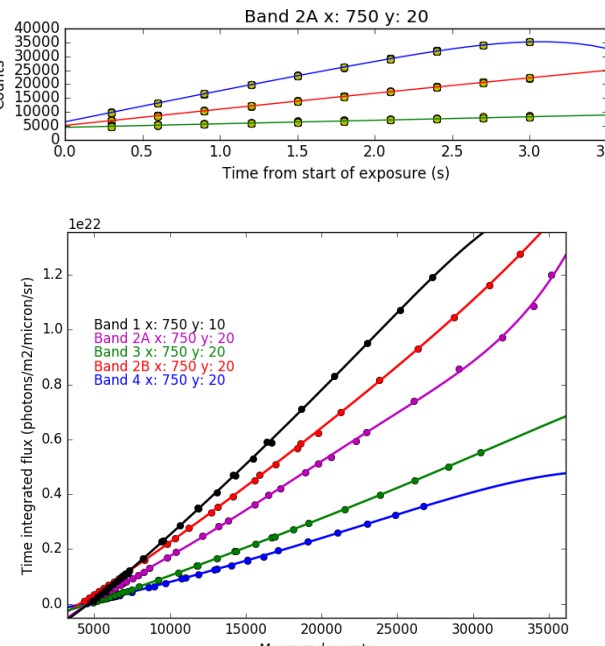

**Figure 12.** Upper panel: measurements of the integrating sphere at three different luminances (blue: high luminance, red: mid luminance, green: low luminance – luminance values are listed in Table 4) from a single Band 2A pixel (pixel number 750 along the x axis, 20 along the y axis within the band). Lower panel: radiometric calibration curve fitted to the integrating sphere observations for a single pixel (pixel numbers listed in the legend) in each of the five GHOST bands.

### 3.6 Summary of GHOST spectrometer performance

Table 5 summarizes the spectral performance of the GHOST spectrometer, specifically the wave-
length range, spectral resolution, and spectral sampling for each band, which we evaluate using the
dispersion functions obtained in Sect. 3.3.

The spectral resolution varies slightly across each band, as the wavelength does not increase lin-
early with pixel number. We estimate the range of spectral resolutions for each band by taking the
narrowest and widest full width half maxima (FWHM) in pixels, obtained through the asymmetric
Gaussian fits to the emission lines, and multiplying them by the spectral sampling (the gradient of
the dispersion function, $d\lambda/dp$ where $p$ is the pixel number) evaluated at the locations of those line
centres. Band 2B has a slightly higher spectral resolution than Band 2A (around $0.19\,\mathrm{nm}$ resolution
compared with around $0.23\,\mathrm{nm}$), though unlike Band 2A its spectral range does not extend as far as
the $CH_4$ Q-branch at $1667\,\mathrm{nm}$.





**Table 5.** Spectral performance of the GHOST spectrometer, as determined from the wavelength calibration measurements described in Sect. 3.3. The ranges of spectral resolution and sampling are obtained from the minimum and maximum fitted full width half maxima of the emission lines measured for each band.

| Band | Lower $\lambda$ (nm) | Upper $\lambda$ (nm) | Range (nm) | Resolution (nm) | Sampling (pixels) |
|------|---------------------|---------------------|------------|-----------------|-------------------|
| 1    | 1237.67 | 1296.26 | 58.59  | $0.107 - 0.148$ | $3.675 - 5.005$ |
| 2A   | 1594.03 | 1671.02 | 76.99  | $0.219 - 0.246$ | $5.825 - 6.545$ |
| 2B   | 1588.75 | 1664.97 | 76.22  | $0.179 - 0.202$ | $4.746 - 5.276$ |
| 3    | 1993.92 | 2089.27 | 95.35  | $0.258 - 0.264$ | $5.394 - 5.577$ |
| 4    | 2270.11 | 2378.73 | 108.62 | $0.250 - 0.280$ | $4.630 - 5.126$ |

**Table 6.** Radiometric performance of the GHOST spectrometer, as determined from the calibration measurements described in Sect. 3.5. The reference radiances are in units of $\mathrm{photons\, s^{-1} m^{-2} \mu m^{-1} sr^{-1}}$, and are typical of the spectral radiance levels observed during the Global Hawk flights (e.g. see Fig. 16).

| Band | Reference radiance | Minimum SNR | Mean SNR | Maximum SNR |
|------|--------------------|-------------|----------|-------------|
| 1    | $2.5 \times 10^{20}$ | 51.3  | **142.2** | 213.3 |
| 2A   | $1.5 \times 10^{20}$ | 244.0 | **367.9** | 487.9 |
| 2B   | $1.5 \times 10^{20}$ | 117.7 | **308.2** | 415.5 |
| 3    | $4.0 \times 10^{19}$ | 138.5 | **211.1** | 314.5 |
| 4    | $2.5 \times 10^{19}$ | 122.0 | **193.9** | 253.2 |

355 GHOST's radiometric performance is evaluated by using the signal to noise ratio (SNR). We evaluate the SNR on a pixel-by-pixel basis by fitting fourth order polynomials to both the mean measured signal and the standard deviation of a combined dataset comprising the high and low luminance sets of measurements (see Table 4) for each active pixel, as a function of the input radiative flux. The ratio between these two polynomials then gives the SNR as a function of the observed radiative flux.

360 The active pixels in each column are finally averaged to obtain the SNR spectrum as a function of wavelength.

 Figure 13 shows the estimated SNR for reference input radiances and integration times representative of the observing during the Global Hawk flights, whilst Table 6 shows the mean, minimum and maximum SNR for each band under these assumptions. Bands 2A and 2B have the highest SNRs of

365 the five GHOST bands with mean values of about 370 and 310 respectively. The other bands have lower SNR values, but are still sufficiently high (around 140 to 210) to be useful for trace gas column retrievals. The SNR spectrum is also used as an input by the retrieval algorithm, which requires an estimate of the uncertainty in the measured spectrum at each wavelength (see Sect. 5.1).




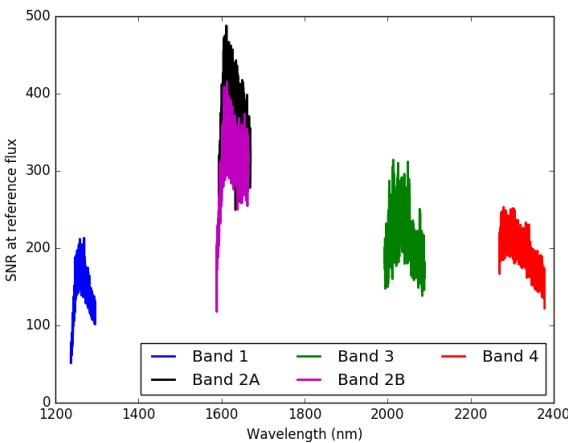

**Figure 13.** Estimate of GHOST SNR for each band derived from integrating sphere measurements, assuming radiances typical of Global Hawk flight observations (Table 6). See text in Sect. 3.6 for details.

## 4 GHOST flights on board the NASA Global Hawk

In February and March 2015, the GHOST instrument was installed and flown on the NASA Global Hawk aircraft (Naftel, 2014) based at NASA Armstrong Flight Research Centre in Edwards, California. The Global Hawk is an unmanned aircraft which is designed for high altitude (up to 20 km) and long endurance flight, and flown remotely by pilots based on the ground at NASA Armstrong. These characteristics make the Global Hawk platform well suited for atmospheric science applica-

tions which demand observations on a regional (as opposed to local) scale from altitudes well above the troposphere.

GHOST flew on the Global Hawk as part of a series of flights jointly supported by the NERC Co-ordinated Airborne Studies of the Tropics (CAST, Harris et al., 2017) and the NASA Airborne Tropical Tropopause Experiment (ATTREX, Jensen et al., 2017) projects. In this section we describe

the installation of GHOST onto the Global Hawk, and provide an overview of the three Global Hawk flights which took place during the joint CAST-ATTREX deployment in February and March 2015.

### 4.1 Installation and operation of GHOST on board the Global Hawk

Following initial verification, testing and calibration at the UK Astronomy Technology Centre, the GHOST instrument was shipped to the NASA Armstrong site (California, USA) in early January

2015. Prior to its integration onto the Global Hawk aircraft, GHOST was required to undergo three major testing periods; electronics and communications, representative environment (pressure and temperature), and mechanical stability (vibrational). These tests were performed and checked by



qualified NASA engineers, with assistance from the GHOST team, as a prerequisite to the instrument receiving permission to fly; NASA has the ability to self-qualify the aircraft and instruments it flies.

Electronics and communications testing was performed both on a lab bench using a test rig simulating the Global Hawk's communications channels, and also on the airfield concourse using the VHF (Very High Frequency) and X-Band communications systems once the instrument was installed on the aircraft. These tests ensured that the power drawn by the system did not exceed limits, that valid log packets were transmitted during operation, that all communications were correct (ports

used and packet contents) and that, *in extremis*, the system could be remotely power cycled and return to operational mode. The communications system uses two basic protocols; a high rate system (over X-band) that provides TCP/IP (Internet Protocol) communications as one would expect on the ground, and a low rate system (over VHF) that provides very limited, multicast-UDP communications. The specifications for instruments to fly on the Global Hawk requires that full control must be

possible using only this system, the data transmission rate for which can be as low as a few hundred bytes every 10 seconds.

   The pressure and temperature testing was performed in an environmental chamber designed to simulate operation under a wide range of temperatures and pressures. The entire GHOST system was installed on the floor of the chamber, then put through a series of pressure and temperature

profiles, with pressures ranging from $1013\,hPa$ down to $56\,hPa$ (corresponding to altitudes from sea level up to $20\,km$) and temperatures from $328\,K$ to $208\,K$. The tests lasted up to four hours and some of the profiles were extremely rapid, demonstrating the instrument's ability to survive a rapid descent from $20\,km$ to near sea level in a few minutes. During these tests the instrument was operational and was tested to ensure that its operational power stayed within limits, with the heaters

and heat exchanger in the ATR being the main power consuming components of the system.

   The vibrational test was performed on each of three orthogonal axes (two horizontal and one vertical) but, owing to the size and weight of the instrument, the ATR, TAM and Spectrometer Module units were tested separately. The testing was again performed with the system operational, although the array was not in an operational mode and was not powered up until the vibration test was

complete. The vibrational specifications used were based on scenarios where forces up to several g were applied, despite the Global Hawk typically operating in a very stable and benign environment. The exact values for the pressure, temperature and vibrational loading are solely for NASA's use and are not publicly available.

   Once all the tests were completed the three sections of GHOST were integrated into the lower in-

strument bay of the Global Hawk (Figure 1) on a specially designed pallet. A final Combined System Test (CST) on the ground with all instruments operational confirmed that GHOST was flight ready. At 15:03 UTC on February 26[th] the Global Hawk took off and embarked on a range flight, climbing to $19\,km$ and then following a race track circuit to the north of Edwards Air Force Base for 6.6 hours, landing at 21:38 UTC (Figure 14). The GHOST system operated continuously and, since the high



rate data communications was available for the majority of the flight, data could be transferred back
in near real time for assessment. The flight demonstrated that all systems were functioning correctly,
the gimbal in the TAM was tracking the solar glint spot (or remaining at nadir) as commanded and
the spectrometer output was stable and consistent with expectations. Subsequently, the header data
from this and subsequent science runs demonstrated that the internal temperatures (the spectrometer

and the array) were stable to better than $0.1\,K$ of the set point, and that the liquid cryogen cooling
system could keep the spectrograph cold for a 20+ hour flight.

**4.2 CAST-ATTREX flights during February and March 2015**

During February and March 2015, the Global Hawk flew on three occasions as part of the CAST-
ATTREX campaign. These comprised a range flight over the desert neighbouring Edwards in South-

ern California (a test of all of the instrument and communications systems on board the Global
Hawk prior to any long range flying, as described in Sect. 4.1), and two science flights over the
Pacific Ocean. These flights are summarised briefly in Table 7, whilst the flight paths are shown in
Fig. 14.

The goal of the ATTREX project was to investigate the physical processes occurring in the tropical

tropopause layer (TTL, Jensen et al., 2017), located between 13 and 19 km in altitude. The project
exploited the unique capabilities of the Global Hawk platform to make observations of stratospheric
humidity and composition. This is reflected in the instrument payload that flew with GHOST on
board the Global Hawk which included *in situ* sensors for water vapor, ice crystals, and ozone, as
well as an air sampling system for post-flight laboratory analysis of chemical composition.

The CAST project (Harris et al., 2017) had similar objectives, looking at atmospheric composi-
tion and structure in the tropics. The 2015 Global Hawk flights, operated jointly with ATTREX, met
the project objectives to demonstrate two new technologies for airborne atmospheric measurements:
GHOST, and an ice crystal imaging probe called AIITS (the Aerosol-Ice-Interface Transition Spec-
trometer, Stopford et al., 2015). The two CAST instruments therefore had opposing requirements

for atmospheric conditions during flight, as GHOST requires the sky to be as cloud-free as possible
for optimal results, whereas the AIITS team would benefit from flying through cloud as much as
possible to fully test their instrument. The two science flights listed in Table 7 each address one of
these two requirements, with a large proportion of the 5th March 2015 flight dedicated to vertical
profiling through the TTL.

The flight on March 10th 2015, on the other hand, was tailored towards the requirements of
GHOST (cloud-free skies during daylight hours). One of the flight legs was designed to coincide
both spatially and temporally with an overpass of the NASA OCO-2 (Orbiting Carbon Observatory,
Crisp et al., 2017; Eldering et al., 2017) satellite during cloud-free conditions as shown in Fig. 15,
providing an opportunity for comparison of the two datasets. In Sect. 5 we focus on this segment

of the March 10th flight when presenting the first set of results from the GHOST spectrometer. This



**Table 7.** Summary of Global Hawk flights which took place under the CAST-ATTREX project during February and March 2015.

| Date (UTC) | Takeoff time (UTC/local) | Duration | Description |
|---|---|---|---|
| **2015-02-26** | 1500/0700 | 6.6 hours | Range flight over desert and mountainous regions close to Edwards, CA. |
| **2015-03-05** | 0400/2000 | 21 hours | Science flight over tropical eastern Pacific targeting profiles through the tropical tropopause layer. About 10 hours of cloud-free daylight conditions on the return leg were suitable for GHOST observations. |
| **2015-03-10** | 1700/1000 | 11.5 hours | Science flight over eastern Pacific targeting clear skies for GHOST observations. The southbound flight leg was specifically located and timed to coincide with an OCO-2 overpass, whilst GOSAT also passed over the measurement region around the same time. |

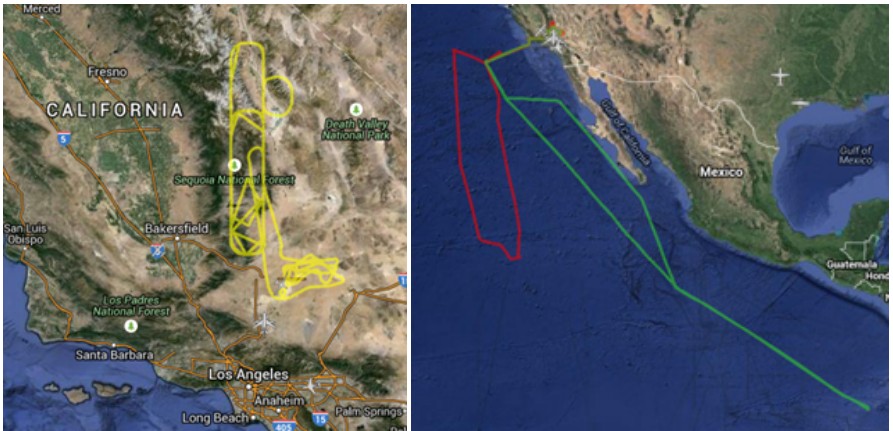

**Figure 14.** CAST-ATTREX Global Hawk flight paths in February and March 2015: 26[th] February 2015 in yellow (range flight); 5[th] March in green (science flight); 10[th] March in red (science flight).

part of the flight also coincided with a GOSAT (Greenhouse Gases Observing SATellite, Kuze et al., 2009) overpass of the region, with the sounding locations shown in yellow in Fig. 15.



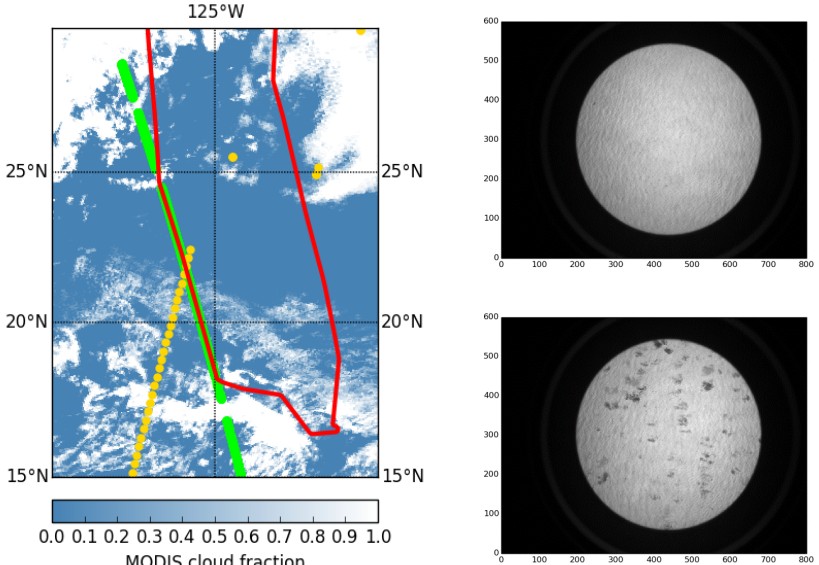

**Figure 15.** Left: 10th March Global Hawk flight path (red) plotted with sample locations of OCO-2 (green) and GOSAT (yellow) soundings. The blue-white colour scale shows the cloud fraction retrieved from MODIS (MODerate resolution Imaging Spectrometer) observations. Right: images from the GHOST visible camera taken during the OCO-2 overpass. The upper image shows cloud free conditions towards the beginning of the flight leg, whilst the lower image shows sparse cloud cover encountered towards the end of the flight leg.

## 5 First GHOST results from the CAST-ATTREX Global Hawk flights

This section describes the first in-flight results obtained from the GHOST spectrometer, taken during the CAST-ATTREX 10th March 2015 Global Hawk flight. The first part of this section outlines the optimal estimation method used to estimate GHG concentrations from the measured spectra, including steps taken to allow for a systematic channeling effect observed in the data, whilst the second part presents the results themselves and assesses their quality.

### 5.1 Retrieval of GHG total column observations from GHOST spectra

The calibrated GHOST spectra (an example of a radiance spectrum recorded during this flight is shown in Fig. 16) have been analysed using the University of Leicester full physics retrieval algorithm. This algorithm uses an iterative retrieval scheme based on Bayesian optimal estimation to estimate a set of atmospheric, surface, and instrument parameters, referred to as the state vector, from the radiance spectra measured by GHOST via calls to a forward model and an inverse method. The forward model describes the physics of the measurement process and relates measured radiances to the state vector. It consists of a radiative transfer (RT) model coupled to a model of





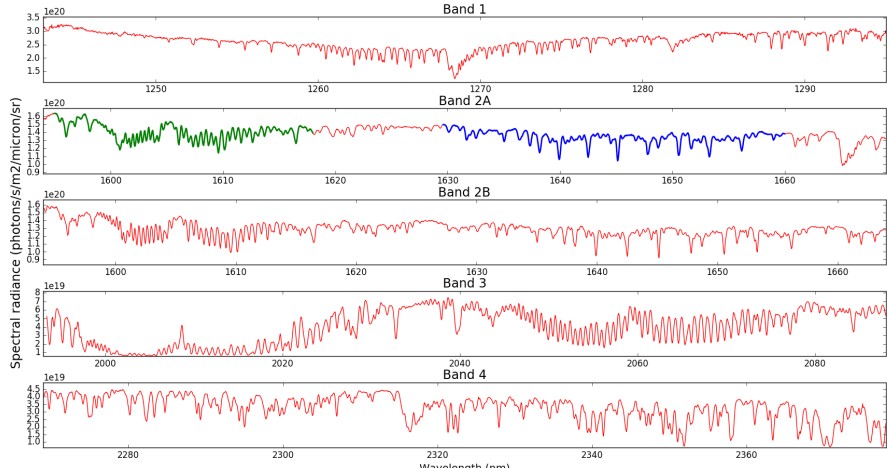

**Figure 16.** Example sun-glint radiance spectra measured by GHOST during the 10$^{th}$ March 2015 Global Hawk flight, with solar zenith angle (SZA) of 36.8°. The fit windows we use in Sect. 5.1 to retrieve $CO_2$ and $CH_4$ from Band 2A are highlighted in green and blue respectively.

the solar spectrum to calculate the monochromatic spectrum of light that originates from the sun, passes through the atmosphere, reflects from the Earth's surface or scatters from molecules or particles in the atmosphere, and is measured by the instrument (Boesch et al., 2006, 2011). We use

output from two re-analysis models as the *a priori* estimate of the atmospheric state: the Copernicus Atmosphere Monitoring Service (CAMS,  Bergamaschi et al., 2007, 2009) for $CH_4$ profiles as a function of pressure, and CarbonTracker (CT2016, Peters et al., 2007, with updates documented at http://carbontracker.noaa.gov) for similar profiles of $CO_2$, $H_2O$, and temperature.

The forward model of the retrieval algorithm employs the LIDORT radiative transfer model (Spurr,

2008), and we use the output of LIDORT at the height of the aircraft in the upward direction. The monochromatic spectrum is then passed through the instrument model where it is convolved with the instrument line shape function to simulate the measured radiances at the appropriate spectral resolution. The inverse method employs the Levenberg-Marquardt modification of the Gauss-Newton method to find the estimate of the state vector with the maximum a posteriori probability, given the

measurement (Connor et al., 2008).

Here, we only present results from a fit to the $CO_2$ and $CH_4$ spectral bands in Band 2A (coloured green and blue respectively in Fig. 16) to infer the ratio of $CO_2$ to $CH_4$, according to the proxy retrieval approach (Parker et al., 2011). The advantage of this approach is the reduced sensitivity to aerosols due to a cancellation of the aerosol effects in the retrieved $CO_2$ and $CH_4$ columns, so sub-

sequently aerosols do not have to be considered for the retrieval. The retrieved state vector elements are listed in Table 8. In future, we plan to also adopt a full-physics retrieval approach where aerosol





**Table 8.** Description of the state vector.

| Quantity | Number of elements | Description |
|---|---|---|
| CH$_4$ | 1 | |
| CO$_2$ | 1 | Scaling factor (no units) applied to a priori vertical profile |
| H$_2$O | 1 | |
| HDO | 1 | |
| Temperature | 1 | Shift (degrees Kelvin) added to a priori vertical profile |
| Albedo | 5 | Coefficients of fourth order polynomial describing surface albedo (no units) as function of pixel number |
| Dispersion | 5 | Coefficients of fourth order polynomial describing wavelength (metres) as function of pixel number |
| Zero level | 1 | Wavelength independent scaling factor (no units), applied to radiance spectrum and then added to approximate effect of straylight in the instrument |
| EOFs | 3 | Wavelength independent scaling factors (no units) applied to empirical orthogonal functions (see text in Sect. 5.1) which are then added to the spectrum to account for spectral systematic errors |

parameters are retrieved together with CO$_2$ by a simultaneous fit to the O$_2$ band (Band 1), the weak CO$_2$ (or CH$_4$) band (Band 2) and the strong CO$_2$ band (Band 3).

During our initial attempts to run the retrieval algorithm on the flight spectra, we noticed a persistent systematic spectral oscillation in the spectral residuals. On revisiting the calibration measurements taken in the laboratory, a similar spectral oscillation was found when taking the difference between spectra measured with and without the TAM included in the optical chain (Appendix A). When light is transmitted by two parallel surfaces, a sinusoidal interference pattern as a function of wavelength is produced. The period of the oscillation at a given wavelength is proportional to the distance between the two surfaces, and the refractive index of the medium between them. Analysis of the period of the oscillation observed in each band suggested that an air gap $0.2\,\mathrm{mm}$ thick could be the source of the interference pattern. Of the optical components within the TAM we identified the depolarizer as the most likely cause, since it includes an air gap of the correct thickness to produce the periodicity of oscillations observed in the GHOST spectra. To resolve this issue, in September 2017 we fitted GHOST with a replacement depolarizer which does not include an air gap, and therefore does not introduce any spectral interference.

In our analyses of flight data recorded before the depolarizer was replaced, we have to take into account this systematic spectral effect. We do this by using the whole GHOST Global Hawk dataset



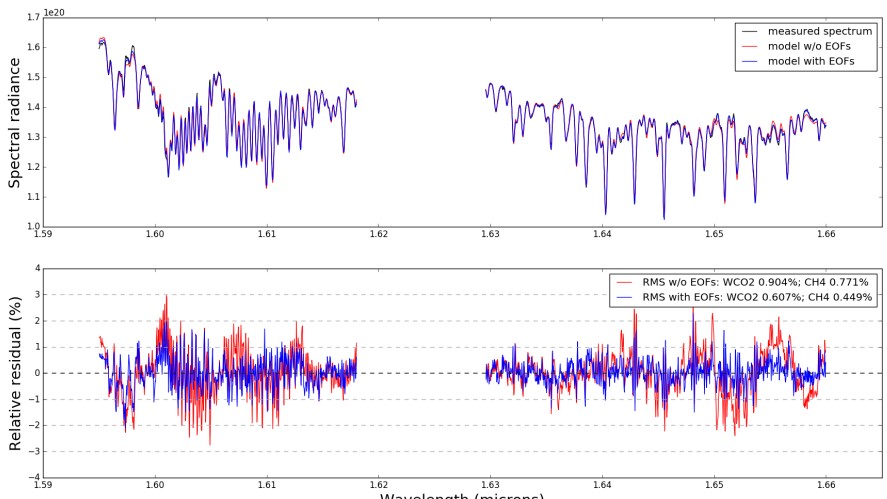

**Figure 17.** Example spectral fit residuals for the Band 2A $CO_2$ (left) and $CH_4$ (right) fit windows, using a sun-glint spectrum measured during the 10[th] March 2015 Global Hawk flight. The measured and optimised forward modelled spectra (both with and without EOFs included in the retrieval algorithm) are plotted in the top panel, whilst the bottom panel shows the spectral fit residuals in each case (modelled – measured) as a percentage of the measured spectrum.

to calculate empirical orthogonal functions (EOFs) describing the variability in the spectral fit resid-

uals obtained during the initial retrieval process (following the method used in OCO-2 retrievals, JPL 2015, to account for systematic errors in the measured spectra). We then run the retrieval algorithm again, this time incorporating into the fit the three EOFs accounting for the most variability in the spectral residuals. Three extra parameters are added to the state vector, which are scaling factors applied to each of the EOFs before they are added to the forward model spectrum. The improvement

in the spectral fit is clearly demonstrated by the spectral residuals shown in Fig. 17, where the red and blue spectra show the residuals obtained without and with EOFs included in the retrieval. In the example shown, including the EOFs in the spectral fit reduces the RMS residuals by 33% and 42% in the $CO_2$ and $CH_4$ fit windows respectively.

We use this retrieval method to obtain our first set of results for $XCO_2$ and $XCH_4$ from a set

of GHOST flight spectra, which were measured during the 10[th] March 2015 Global Hawk flight described in Sect. 4. We focus in particular on the 218 spectra observed from 21:54 UTC onwards during the OCO-2 and GOSAT overpasses. On applying quality filters based on thresholds for the number of algorithm iterations and the $\chi^2$ value of the residual, 175 observations remain. The left panel of Fig. 18 shows how the retrieved $XCO_2$ (green), $XCH_4$ (blue) and ratio (red) values vary

along this leg of the flight, whilst the right panels show how the RMS residual values behave for





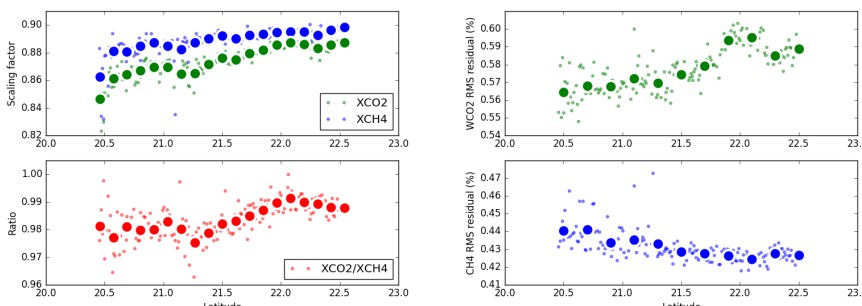

**Figure 18.** Left: scatter plots for the Band 2A retrieved $XCO_2$ (green) and $XCH_4$ (blue) scaling factors, along with the ratios (red), from spectra measured during the OCO-2 overpass of the $10^{th}$ March 2015 Global Hawk flight. Right: scatter plot of RMS residual for the Band 2A $CO_2$ (green) and $CH_4$ (blue) fit windows. The data is averaged into $0.2°$ latitude bins (the average for each bin is indicated by the larger circles).

the quality-filtered retrievals. This metric, along with the $\chi^2$ values (not shown), indicate that the spectral fits of the $CH_4$ window using the OCO retrieval algorithm are superior to those of the $CO_2$ window.

The range of values covered by the ratio between the two scaling factors, as shown by the red data
points in Fig. 18, is of the order of 1.5%. When applied to the prior total column concentration of $CO_2$ via the proxy retrieval method described in Sect. 5.1, this corresponds to a variation in $XCO_2$ along this flight leg of around 6 ppm. This is greater than that seen over the same latitude range in both the OCO-2 observations and the CarbonTracker atmospheric model, which both indicate a spread in $XCO_2$ values of around 1.5 ppm. The variability in the scaling factor ratio is driven more
by the retrieved $CO_2$ scaling factor than that of $CH_4$, as can be seen from the top left panel of Fig. 18. The divergence of the $CO_2$ scaling factor away from the $CH_4$ scaling factor also results in a negative bias compared with the OCO-2 data and the CarbonTracker model output, which varies from 2 ppm up to around 8 ppm along the flight leg. Whilst we are still investigating the underlying cause of this bias, we anticipate that an improved understanding of straylight effects in the instrument through
ongoing laboratory measurements will help us to correct for the observed bias, as will comparison of retrieval results from future GHOST flights against those from ground-based validation sites.

## 6   Summary

GHOST is a new shortwave infrared grating spectrometer, which employs a unique compact design to optimize the science benefits provided (with regards to wavelength ranges covered and spectral
resolution) given the small size of its optical payload. As described in Sect. 2, this has been achieved by adapting advances in optical technology originally designed for astronomy applications, namely



the use of fibre bundles to transfer light from the fore-optics into the spectrometer, and the imaging of multiple spectral bands onto a single detector array with a single diffraction grating. Through the laboratory testing and calibration discussed in Sect. 3, we have demonstrated that the GHOST radio-

metric and spectral performance is suitable for retrieving total column concentrations of greenhouse gases from the measured shortwave infrared spectra.

The CAST-ATTREX Global Hawk flight campaign in January-March 2015 (Section 4) was our first opportunity to operate GHOST on board an aircraft. Following extensive engineering and communications testing at NASA Armstrong, GHOST was successfully integrated into the Global Hawk

aircraft, and flew for the first time on February 26th 2015. During the campaign, GHOST took part in three flights on board the Global Hawk for a total duration of approximately 39 hours, maintaining vacuum and thermal stability throughout up to altitudes of 20 km above sea level. Data from the March 10th 2015 flight are analyzed in Sect. 5, which shows for the first time greenhouse gas retrievals from spectra measured in-flight by GHOST.

This paper has introduced the GHOST instrument, and shown its potential for airborne greenhouse gas remote sensing through initial results from its first science flights. Our future plans for this dataset include applying a full physics retrieval to the measured spectra (as discussed in Sect. 5.1), which will allow for a more like-for-like comparison with the OCO-2 measurements during the 10th March 2015 overpass. In addition to the CAST-ATTREX Global Hawk flights we have also flown

GHOST over the UK on board the NERC Airborne Research Facility (ARF) Dornier 228 aircraft in April and May 2015, targeting emissions hot spots. Future plans include analysis of this dataset, as well as further flights supported by the ARF in Spring 2018. These flights will investigate how multi-angle views of a scene can improve the retrieval accuracy through better representation of scattering effects in the atmosphere, as well as targeting ground-based remote sensing instruments

in the UK which can be used for validation of the GHOST results. Future flights (including those planned for 2018) will benefit from a new depolariser unit which has now been installed in the TAM, eliminating the spectral oscillations that affected previous measurements (see Appendix A). In addition we are working on further calibration measurements prior to these flights, which will include characterization of straylight effects and will enhance our understanding of the instrument, therefore

improving our ability to retrieve GHG concentrations from the measured spectra. GHOST is also suitable for more dedicated inter-comparison studies with greenhouse gas observing satellites, including the Sentinel-5 Precursor which observes $CH_4$ using a similar wavelength range to GHOST's Band 4. We anticipate that GHOST will prove to be a valuable tool for regional and local scale studies of the atmospheric greenhouse gas budget, in addition to acting as an airborne technology

demonstrator for future satellite missions.





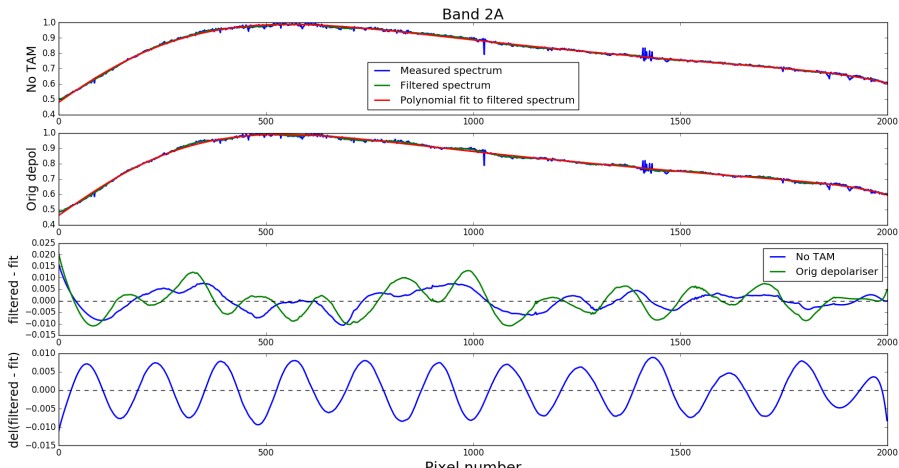

**Figure A1.** Uncalibrated Band 2A measurements of integrating sphere output. From top to bottom: spectrum measured without the TAM in place (blue), which is then noise filtered (green) and approximated by a polynomial (red); same as top panel, but with the TAM in place; difference between noise filtered spectrum and the polynomial fit in each of the two cases; difference between the two residuals shown in the third panel, isolating the effect of the depolariser on the spectrum.

## Appendix A: Observation of depolarizer interference effect in laboratory measurements

The effect of the depolarizer discussed in Sect. 5.1 on GHOST Band 2A measurements is shown in Fig. A1. We perform measurements with the TAM removed from the optical chain by decoupling the optical fibre bundle from the TAM and feeding it directly into the output port of the integrating sphere. This results in observed intensities that are not identical, since the incoming light is no longer being focused onto the fibre bundle by the TAM fore optics. To allow for this, we scale the two spectra such that the maximum value in each is equal to 1. We then apply a noise filter and perform a polynomial fit to isolate the background signal. The bottom panel shows the difference between the two measurements once these parts of the spectra are removed, leaving behind only the spectral effect of the depolariser. Note that these are uncalibrated measurements: the radiometric calibration coefficients obtained from the integrating sphere measurements (Section 3.5) were obtained with the TAM in place, so are only applicable in that configuration.

*Acknowledgements.* The authors would like to thank the team at the NASA Armstrong Flight Research Centre for their assistance during the CAST-ATTREX campaign, particularly Dave Fratello who provided us with invaluable logistical support both before and during the NASA Armstrong deployment. Additional work (beyond that of the co-authors) on the design, mechanical build, software development and deployment of GHOST was provided by STFC-ATC engineers George Davidson, Xiaofeng Gao, Brian Woodward, Brian Wilson, and





Tom Baillie. We also acknowledge contributions towards the optical design of GHOST by STFC-ATC optical engineers Andy Born and Martin Black. The design, manufacture and deployment of GHOST was co-funded

by NERC and STFC through the Co-ordinated Airborne Studies of the Tropics (CAST) project, with grant numbers NE/I030054/1 (lead award), NE/J006211/1, NE/J006238/1 and NE/J006203/1. Subsequent funding from the UK Centre for Earth Observation Instrumentation (CEOI) has supported and enabled further study of GHOST beyond the original CAST project. This research used the SPECTRE and ALICE High Performance Computing Facilities at the University of Leicester. The calibration measurements described in Sect. 3 were

made using facilities and laboratory space at the UK Astronomy Technology Centre (UK-ATC) in Edinburgh, UK, with further equipment and expertise provided by Chris MacLellan of the Natural Environment Research Council Field Spectroscopy Facility (NERC FSF). The retrieval code described in Sect. 5.1, originally developed for satellite measurements from above the atmosphere, was adapted for observations from an aircraft instrument within the atmosphere by Peter Somkuti at the University of Leicester. The MODIS L2 Cloud Prod-

uct data used in Fig. 15 was acquired from the NASA MODIS Adaptive Processing System, located in the Goddard Space Flight Center in Greenbelt, Maryland (http://modis-atmos.gsfc.nasa.gov/MOD06_L2/).



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
