# Peer review of "GreenHouse gas Observations of the Stratosphere and Troposphere (GHOST): an airborne shortwave infrared spectrometer for remote sensing of greenhouse gases"

_Atmospheric Measurement Techniques, 2017_

## Referee Comment (RC1) · Anonymous Referee #1 · 22 May 2018

With great interest I have read the manuscript "GreenHouse gas Observations of the Stratosphere and Troposphere (GHOST): an airborne shortwave infrared spectrometer for remote sensing of trace gases"

In this paper the authors present a new instrument (GHOST) that has been developed for remote sensing of CO2 and CH4 vertical columns. The optical design of the instrument (essentially a pointable telescope coupled to the GHOST spectrometer) is described in detail in the first part of the manuscript. This section is followed by a detailed description of the instrument spectral and radiometric calibration by making use

of different light sources and optical elements (emission lamps, QTH lamps, integrating sphere). The second part of the paper describes the deployment of the GHOST instrument from the NASA Global Hawk aircraft, as part of the CAST-ATTREX campaign (Feb/Mar 2015) and includes a description of the retrieval algorithm which focuses on the CO2 to CH4 ratio in order to reduce the sensitivity to aerosols.

The GHOST instrument is designed for monitoring some of the most relevant atmospheric constituents and it may have an important added value in relation to the validation of current and future CO2 and CH4 satellite data products. The article is wellwritten and thorough in its description of the GHOST instrument and its optical characterization, which is the main subject of this paper. In my view the results obtained during the airborne campaign are important to demonstrate the abilities of the instrument and retrieval algorithm. The data volume is relatively small however, which limits the scope of the first results. I consider it a strong aspect of the study that the authors we able to identify a relation between oscillations in the residuals of the spectral fit and certain optical components in the telescope unit. This shows a thorough understanding of all aspects related to the instrument design and data processing and has lead to an instrument improvement (replacement polariser) that was made after period considered in this paper. Altogether I recommend this paper for publication in AMT, after the following minor comments have been addressed.

p.2,I.25: please add a reference for this requirement in the Paris Agreement.

p.16,I.309: "we interpolate.." Please specify interpolation method (e.g. linear).

p.17, I.341: Considerable non-linear behaviour can also be observed below 26000 counts, e.g. for band 1. In practice, do you apply the full calibration curve to convert counts to time-integrated flux, or do you use linear coefficients to approximate these curves?

p.25, I.485: I suggest to replace "output of LIDORT at the height of the aircraft in the upward direction" with "upwelling radiance at the height of the aircraft"
p.24, I.492: Please mention - in a few sentences - the most essential assumptions underlying the proxy retrieval approach.

p.27, I.527: Please be more specific about the applied thresholds. How were these determined? How much data was discarded?

p.28, I.534-I.546: A different spread in XCO2 is observed in the GHOST airborne observations compared to the CarbonTracker model and OCO-2 satellite observations. Please discuss the possible impact of the assumptions underlying the proxy retrieval approach on this observed discrepancy: how robust are these assumptions in relation to the magnitude of the observed discrepancy (6ppm versus 1.5ppm)?

Please also discuss the possible impact of differences in spatial resolution / representativeness on the observed discrepancy.

---

## Referee Comment (RC2) · Anonymous Referee #2 · 23 May 2018

The manuscript AMP 2017-274 "GreenHouse gas Observations of the Stratosphere and Troposphere (GHOST): an airborne shortwave infrared spectrometer for remote sensing of greenhouse gases" represents a substantial contribution to scientific progress within the scope of Atmospheric Measurement Techniques because it introduces a new instrument. The document is well written and properly structured. My comments and suggestions for improvement of the text and figures are given below. I recommend this paper for publication in AMT after consideration for the comments.

The paper address relevant scientific questions within the scope of AMT because it

introduces a new instrument "GHOST" that can be mounted in an airplane or UAV for spectroscopic measurement of greenhouse gases (GHG) in the portion of the Earth atmosphere underneath the flight track. A broad and well written introduction with a useful overview of relevant ground based, airborne and satellite GHG instrumentation is given. The GHOST instrument aims to fill the niche between existing ground observations on the local scale and satellite measurements on regional to global scale. Once mature enough, it could potentially be used for validation of satellite measurements.

The GHOST instrument is a fiber-fed spectrometer. Multiple orders of a single grating are used to combine high spectral resolution with compactness of the instrument. These methods are known from astronomy instrumentation. The GHOST instrument realization shows that airborne/AUV applications can be a valuable demonstration environment for remote sensing instruments, this is the substantial novelty in this work. There is a promise of useful data for validation of ground based or satellite. For this purpose the instrument, its calibration and data reduction would need to become more mature.

Scientific methods are clearly outlined and referenced. The GHOST instrument is introduced in a descriptive way. A more architectural description would be preferred to help the reader better understand the layout and working of the instrument.

The main result is the build-up of the instrument. A more detailed description of the optical layout supported by block diagrams or schematic of the optical path would largely improve the scientific value. A discussion of choices/trade-offs in the optical design and a description of the specifications and tolerances of the optical components is lacking.

Some minor point to be considered; the detector performance is characterized at 80 K but operated at a higher temperature, please discuss if the difference has an impact on the performance; The instrument employs two bands 2. A comparison of simultaneous measurements for the two bands could give insight in the instrument performance; The horizontal scale for bands 2A and 2B in figure 16 is different. Identical scales would be

beneficial for comparison.

In conclusion, the presented instrument and its first application are a substantial and valuable contribution to the scientific field of atmospheric remote sensing. The innovation to combine multiple grating orders on a single detector is very interesting for its compactness. The paper could be significantly improved if the architectural design of the optical instrument would be given in view of the operational performance (design trade-offs, effect of stray light, separation of diffraction orders, suppression of spurious orders). Such treatment would be crucial to assess the wider applicability of the presented instrument and to assess future inter-comparison/validation studies.

---

## Author Comment (AC2) · 31 Jul 2018

The authors would like to thank Anonymous Referee #2 for their constructive and helpful feedback on the AMT Discussion Paper 'GreenHouse gas Observations of the Stratosphere and Troposphere (GHOST): an airborne shortwave infrared spectrometer for remote sensing greenhouse gases' by Humpage et al. Here, we address each of the referee's comments in turn and outline how we have used them to improve the manuscript.

[Figure]

'The GHOST instrument is introduced in a descriptive way. A more architectural description would be preferred to help the reader better understand the layout and working of the instrument.'

- We have added a block diagram (Fig. 1) to the manuscript to provide an architectural overview of how the three components of GHOST work together.

'A more detailed description of the optical layout supported by block diagrams or schematic of the optical path would largely improve the scientific value.'

- We have added schematic diagrams of the optical path inside the TAM (Fig. 2) and the spectrometer (Fig. 3) to clarify the concept behind the optical design.

'A discussion of choices/trade-offs in the optical design and a description of the specifications and tolerances of the optical components is lacking.'

- We have added some discussion of the specifications and tolerances in the optical design of the TAM (Sect. 2.2 of the manuscript) and the Spectrometer Module (Sect. 2.3) to address this comment.

'Some minor point to be considered; the detector performance is characterized at 80 K but operated at a higher temperature, please discuss if the difference has an impact on the performance;'

- We have added the following text to the manuscript in Sect. 3.2 to address this point: 'The measured dark current values are higher than the 0.25 counts/s that would be achieved if the detector were operated at liquid nitrogen temperature (80 K). However we consider it more advantageous to keep the detector slightly warmer but at a stable temperature (and known dark current, which can then be subtracted more easily), than to minimize the dark current at the expense of temperature stability.'

'The instrument employs two bands 2. A comparison of simultaneous measurements for the two bands could give insight in the instrument performance; The horizontal scale for bands 2A and 2B in figure 16 is different. Identical scales would be beneficial for

comparison.'

- We have updated the original Figure 16 (Fig. 4 here) such that both the wavelength and the spectral radiance scales for Bands 2A and 2B are now identical. Whilst the spectral performance is different as expected (because of the different spectral sampling and resolution achieved for each band), it is worth noting that the radiometric calibration (as projected onto the spectral radiance scale) produces different results for the two bands. A possible explanation for this is that straylight, which is not accounted for here, affects the two bands differently. We are currently investigating the effects of straylight using new laboratory measurements taken since this work was submitted for publication, and have updated the text in the manuscript (Section 6) to refer to this and link it to the updated figure.

───────────────────────────────

[Figure]

[Figure]

**Fig. 1.** Block diagram showing the three main components of the GHOST instrument and how they interact with one another.

Fibre End
Lenses
Depolarising Unit
Altitude Axis
Pupil Stop
Fold 2
Fold 1
Off-axis-
parabolic Mirror
Dome
Azimuth Axis

Fibre bundle
end

9.9mm

1mm

8.8mm

14.66mm

Image stop

Two identical lenses

Material: infrared grade fused silica

Both surfaces of both lenses convex with
radius of curvature 19mm

Centre thickness 2.5mm

Clear aperture needed: 7mm diameter.
Picture here assumes 9mm diameter
component, but might be easier to handle if
slightly larger

**Fig. 2.** Optical path diagram showing how the optical components of the TAM direct the observed light into the fibre bundle.

**Spectrometer optical bench**

Grating

Blocking filter

Condensing lens

MCT detector array

Fibre optic feed

Control electronics

**Fig. 3.** Block diagram showing a simple overview of the GHOST spectrometer optical layout.

[Figure]

**Fig. 4.** Example sun-glint radiance spectra measured by GHOST during the 10th March 2015 Global Hawk flight, now with matching scales for Bands 2A and 2B.

---

## Editor Comment (EC1) · U. Platt (Editor) · 2 Aug 2018

The comments of two anonymous reviewers (#1 and #2) were quite positive and both reviewers found the manuscript suitable for AMT after some suggested revisions.

When reading the authors' responses (including the intended changes to the text of the manuscript) to the comments of reviewers #1 and #2 I feel that the authors sufficiently well addressed the reviewers' comments, therefore I would encourage the authors to prepare a revised version of the manuscript.

[Figure]

In addition to the reviewers' comments I would encourage the authors to further clarify the manuscript by adding the following information:

Detector dark current: The dark current is given as 0.25 electrons per second and pixel (section 2.2) and as about 5 counts (Table 2). It should be clarified at which temperature the dark current figures in Table 2 were measured (80K or 98K ?). In any case – as requested by reviewer #2 – the dark current at 98 K should be given. The standard deviation of the dark current in Table 2 appears very high, this needs explanation.

Digital data: The relationship between electrons and 'counts' should be given. Also, the max. exposure in counts (probably 65000) is not explicitly stated, this information should be added in order to allow the reader to judge the fraction of dark current to the signal.

I am looking forward to reading the revised manuscript. with very best regards Ulrich Platt

---

## Author Response (AR1)

**GreenHouse gas Observations of the Stratosphere and Troposphere (GHOST): an airborne shortwave infrared spectrometer for remote sensing of greenhouse gases**

Neil Humpage[1], Hartmut Boesch[1,2], Paul I. Palmer[3,4], Andy Vick[5,6], Phil Parr-Burman[5], Martyn Wells[5], David Pearson[5], Jonathan Strachan[5], and Naidu Bezawada[5,7]

1. Earth Observation Science, Department of Physics and Astronomy, University of Leicester, Leicester, UK
2. National Centre for Earth Observation, Leicester, UK
3. School of GeoSciences, University of Edinburgh, Edinburgh, UK
4. National Centre for Earth Observation, Edinburgh, UK
5. Science and Technology Facilities Council, UK Astronomy Technology Centre, Edinburgh, UK
6. Now at Science and Technology Facilities Council, Rutherford Appleton Laboratory, Oxfordshire, UK
7. Now at European Southern Observatory, Garching, Germany

**Author's response to RC1, 2018-07-31 (blue text in marked-up manuscript):**

The authors would like to thank Anonymous Referee #1 for their constructive and helpful comments on 'GreenHouse gas Observations of the Stratosphere and Troposphere (GHOST): an airborne shortwave infrared spectrometer for remote sensing of greenhouse gases' by Humpage et al. Here we respond to the specific comments in turn:

p.2, l.25: please add a reference for this requirement in the Paris Agreement.

- We have amended the reference to the Paris Agreement in the text to include the specific sections relevant to this requirement.

p.16, l.309: "we interpolate.." Please specify interpolation method (e.g. linear).

- We use a two step interpolation: firstly, a linear interpolation is used to determine the line shape centred on each wavelength pixel; then, each pixel-specific line shape is interpolated in log(delta-wavelength) space to determine how it maps onto the neighbouring pixels. We have now added this to the manuscript.

p.17, l.341: Considerable non-linear behaviour can also be observed below 26000 counts, e.g. for band 1. In practice, do you apply the full calibration curve to convert counts to time-integrated flux, or do you use linear coefficients to approximate these curves?

- We apply a fifth-order polynomial fit to the full calibration curves, and then use these coefficients to convert our flight spectra from counts into time-integrated flux. We have amended the text slightly to make this more clear to the reader.

p.25, l.485: I suggest to replace "output of LIDORT at the height of the aircraft in the upward direction" with "upwelling radiance at the height of the aircraft"

- We agree with the referee, and have updated the wording of the text as suggested.

p.24, l.492: Please mention - in a few sentences - the most essential assumptions underlying the proxy retrieval approach.

- We have added a couple of sentences outlining the key assumptions – specifically, we assume that very similar distributions of light paths contribute to the observed spectra at the absorption wavelengths of both the gas of interest and its proxy for the total air column, and that our

measurement exhibits equivalent sensitivity to both gases at the heights where the greatest deviations from the expected light path occur as a result of scattering). We have also added a couple of references for further detail if required.

p.27, l.527: Please be more specific about the applied thresholds. How were these determined? How much data was discarded?

- We apply thresholds on the number of algorithm iterations required to converge on a solution (must be less than 7, based on using histograms of the data to identify outliers), and on the chi squared goodness-of-fit statistic (must be greater than 1, as results less than 1 are unphysical). This results in 43 out of 218 spectra along this leg of the flight being discarded. We have updated the text to describe the thresholds, including a reference justifying use of the chi-squared threshold.

p.28, l.534-l.546: A different spread in XCO2 is observed in the GHOST airborne observations compared to the CarbonTracker model and OCO-2 satellite observations. Please discuss the possible impact of the assumptions underlying the proxy retrieval approach on this observed discrepancy: how robust are these assumptions in relation to the magnitude of the observed discrepancy (6ppm versus 1.5ppm)?

- We have added the following text to the manuscript to address this comment: 'When using the proxy retrieval method, we are assuming that the two spectral ranges used are close enough in wavelength space that any systematic errors are eliminated when taking the ratio. However, this would not work (and could subsequently introduce the observed divergence in scaling factors obtained from the two spectral ranges used) in the presence of an instrumental error or effect which has sufficient wavelength dependence within the wavelength ranges observed by GHOST. Examples of these include unaccounted for straylight incident on that part of the detector, or a wavelength dependent error in the radiometric calibration.'

Please also discuss the possible impact of differences in spatial resolution / representativeness on the observed discrepancy.

- We have tried to address this comment by adding the following text to the manuscript: 'Given that OCO-2 makes observations at a higher spatial density than GHOST (because of its eight pixel cross-track imaging capability), we would expect OCO-2 data to be more representative of the variability in XCO2 along the overpass compared with GHOST. We can infer from this that the change in discrepancy between the two datasets along the flight track is unlikely to be a result of spatial changes in XCO2 that OCO-2 might be missing as a result of mismatches in spatial resolution and location.'

**Author's response to RC2, 2018-07-31 (red text in marked-up manuscript):**

The authors would like to thank Anonymous Referee #2 for their constructive and helpful feedback on the AMT Discussion Paper 'GreenHouse gas Observations of the Stratosphere and Troposphere (GHOST): an airborne shortwave infrared spectrometer for remote sensing greenhouse gases' by Humpage et al. Here, we address each of the referee's comments in turn and outline how we have used them to improve the manuscript.

'The GHOST instrument is introduced in a descriptive way. A more architectural description would be preferred to help the reader better understand the layout and working of the instrument.'

- We have added a block diagram (Fig. 1) to the manuscript to provide an architectural overview of how the three components of GHOST work together.

[Figure]

*Figure 1: Block diagram showing the three main components of the GHOST instrument and how they interact with one another.*

'A more detailed description of the optical layout supported by block diagrams or schematic of the optical path would largely improve the scientific value.'

- We have added schematic diagrams of the optical path inside the TAM (Fig. 2) and the spectrometer (Fig. 3) to clarify the concept behind the optical design.

[Figure]

*Figure 2: Optical path diagram showing how the optical components of the TAM direct the observed light into the fibre bundle.*

[Figure]

*Figure 3: Block diagram showing a simple overview of the GHOST spectrometer optical layout.*

'A discussion of choices/trade-offs in the optical design and a description of the specifications and tolerances of the optical components is lacking.'

- We have added some discussion of the specifications and tolerances in the optical design of the TAM (Sect. 2.2 of the manuscript) and the Spectrometer Module (Sect. 2.3) to address this comment.

'Some minor point to be considered; the detector performance is characterized at 80 K but operated at a higher temperature, please discuss if the difference has an impact on the performance;'

- We have added the following text to the manuscript in Sect. 3.2 to address this point: 'The measured dark current values are higher than the 0.25 counts/s that would be achieved if the detector were operated at liquid nitrogen temperature (80 K). However we consider it more advantageous to keep the detector slightly warmer but at a stable temperature (and known dark current, which can then be subtracted more easily), than to minimize the dark current at the expense of temperature stability.'

'The instrument employs two bands 2. A comparison of simultaneous measurements for the two bands could give insight in the instrument performance; The horizontal scale for bands 2A and 2B in figure 16 is different. Identical scales would be beneficial for comparison.'

- We have updated the original Figure 16 (Fig. 4 here) such that both the wavelength and the spectral radiance scales for Bands 2A and 2B are now identical. Whilst the spectral performance is different as expected (because of the different spectral sampling and resolution achieved for each band), it is worth noting that the radiometric calibration (as projected onto the spectral radiance scale) produces different results for the two bands. A possible explanation for this is that straylight, which is not accounted for here, affects the two bands differently. We are currently investigating the effects of straylight using new laboratory measurements taken since this work was submitted for publication, and have updated the text in the manuscript (Section 6) to refer to this and link it to the updated figure.

[Figure]

Figure 4: Example sun-glint radiance spectra measured by GHOST during the 10th March 2015 Global Hawk flight, now with matching scales for Bands 2A and 2B.

Manuscript prepared for Atmos. Meas. Tech.
with version 2014/09/16 7.15 Copernicus papers of the LaTeX class copernicus.cls.
Date: 1 August 2018

[revised manuscript text omitted]

**2.1 The Target Acquisition Module (TAM)**

125 The function of the TAM is to direct sunlight which has passed down through the atmosphere, and then been reflected back upwards by the Earth's surface, onto an optical fibre bundle which transfers the light into the spectrometer module. To achieve this, the TAM has to be able to acquire light from any angle of rotation and up to $70°$ away from nadir, depending on the solar zenith angle. A custom off-axis telescope arrangement is used, mounted to a custom Newmark GM-6 gimbal (see Fig. 3).

130 A fold mirror reflects the light path along the elevation axis of the gimbal, through a de-polarising unit, and onto a second fold mirror that directs the light path along the rotation axis (see Fig. 4)[RC2]. Mounted centrally along this axis are the pre-fibre optics (a field stop and lenses) and the fibre bundle mounting point. A field stop of diameter 2.6 mm is used to define the range of angles from which light will be accepted into the fibres, such that it observes reflected light from a surface footprint

135 $1°$ in diameter.[RC2] The optics create a pupil image on the end of the fibre bundle such that the light is both concentrated onto the fibres and evenly distributed between them. The field stop defines the range of angles that will be accepted into the fibres, which is set to an angle of $\pm 6.67°$ (numerical aperture NA = 0.11)[RC2]. In order to avoid misalignment between the main optical ray defined by these apertures and the mechanical axis of the gimbal (which would introduce a shift in pointing

140 angle which varies with gimbal position), we required that the end of the fibre bundle, the coupling lens, and the field stop aperture should be centred on the rotation axis of the gimbal to within $50\,\mu m$. In the focus direction (along the optical axis) they should be positioned to within an accuracy of

[Figure]

**Figure 4.** Optical path diagram showing how the optical components of the TAM direct the observed light into the optical fibre.[RC2]

200 μm. The intersection of the two mechanical axes of the gimbal must lie within 50 μm of the surface of fold mirror 2 (see Fig. 4), whilst the angle of this mirror must also be correct to within 0.1°.[RC2]

By using a design which employs movable optics to transfer the observed light onto a static fibre bundle, we minimize the potential for deterioration of the measured signal caused by movement and bending of the fibres. The gimbal also carries a wide field pan-chromatic camera with a field of view extending beyond that of the spectrometer input. The camera is used to help interpret the science data, e.g. by identifying when the spectrometer field of view is cloud-free.

The light enters the TAM through a dome whose centre of curvature has been placed at the centre of curvature of the gimbal, to reduce differential aberrations across the gimbal range. Originally the whole bottom face of the TAM vessel was designed to be external to the aircraft, since the vessel was intended as an environmentally benign environment for the gimbal and the optics, with internal heaters and the potential for dry gas flushing. The TAM is pressurized to 3 PSI above local ambient conditions to minimize the diffusion of water vapour into the optical system. The vessel also included a number of fiducial measurement points which were used to ascertain alignment and location of the optics, both in the lab and on the aircraft. The final installation was more enclosed and left only the dome itself protruding from the underside of the Global Hawk.

[Figure]

**Figure 5.** GHOST spectrometer module showing the front end compartment, the main spectrometer optics area and the liquid nitrogen cryogenic vessel. The cylinder containing these components is 1064 mm long and 440 mm in diameter. Taking into account the fittings external to the cylinder, the space occupied is 1406 mm long, 758 mm wide, and 440 mm tall.

The optical fibre bundle which passes the received light onto the spectrometer module is made up of 35 multi-mode fibres, each of $0.365\,\text{mm}$ core, encased in a stainless steel outer sheath for protection. At the spectrometer end the bundle of 35 fibres is split into five smaller bundles (one for each spectral band, see Sect. 2.2) comprising seven fibres each, with the choice of fibres kept as spatially random as possible to ensure that a similar sampling of the aperture is passed into each bundle. The light from each of the bundles is transmitted into the (cold) spectrometer through a window and an order sorting filter, which minimises the out of band light, and is finally coupled into a cold fibre bundle.

**2.2 The Spectrometer Module**

Inside the spectrometer module there are three major sections: an end section that contains the optical entry points, cold fibres and the detector box; a middle section containing the spectrometer optics; and a section where the cryogen tank is situated (see Fig. 5). All of the spectrometer module internal components are cooled to $80\,\text{K}$, which has the effect of reducing the thermal background, the dark current on the detector, and any thermo-mechanical instabilities. The decision to use liquid nitrogen for cooling rather than a closed cycle cooler is based primarily on the need to minimise vibrations, both for this instrument and for other instruments mounted on the same aircraft. Cryogenic cooling has the additional benefits of requiring no electrical power, being very reliable (compared with using mechanical cooling which relies on moving components), and using little extra space for external electronics.

[Figure]

**Figure 6.** Block diagram showing a simple overview of the GHOST spectrometer optical layout.[RC2]

We summarise the spectrometer optical design schematically in Figure 6.[RC2] Observed light is
180  brought into the spectrometer optics using cold fibre bundles as mentioned in Sect. 2.1. The optical
part of the spectrometer module comprises four major components: the inlet slits, fold mirror, camera
mirror, and grating (see Fig. 7). The f/5 focused beam from the input slits is reflected by the fold
mirror on to one side of the curved camera mirror. The collimated beam is then reflected again by
the fold mirror onto the grating, which disperses the light from each band into a number of spectrally
185  resolved images, each produced by a different order of diffraction. The dispersed light is reflected
again by the fold mirror on to the other side of the camera mirror, after which the converging beam is
reflected once more by the fold mirror and passed through cylindrical output lenses that concentrate
the spectral images onto the detector. The major innovation in this design is the use of multiple
grating orders; through choice of orders and positioning of the input slits, multiple high resolution
190  spectral bands can be selected from a wide overall wavelength range and conveniently placed onto
the output plane. In the GHOST spectrometer we use a 60 grooves $\mathrm{mm}^{-1}$ blazed grating supplied
by Richardson Gratings, with a blaze angle of $28.7°$ corresponding to a blaze wavelength of $16\,\mu\mathrm{m}$.
For spectral bands 1 through to 4 (Table 1) we use diffraction orders 13, 10, 8, and 7 respectively.
The grating and orders were chosen to minimise the spread of input slit offsets (along the dispersion
195  direction at the entrance to the spectrometer) required to image all of the bands onto the detector,
using band 2 as the reference point. The spread was calculated for diffraction orders up to 18, with
the result that using diffraction order $n = 10$ for band 2 minimised the need to shift the input slits
for the other bands. We then selected an appropriate grating by requiring that the blaze wavelength
be as close to $n\lambda$ for band 2 as possible.[RC2]

[Figure]

[revised manuscript text omitted]
 2. These values are higher than the $0.25\,\mathrm{counts\,s^{-1}}$ that would be achieved if the detector were operated at liquid nitrogen temperature (80 K, see Sect. 2.2). However we consider it more advantageous to keep the detector slightly warmer but at a stable temperature (and known dark current, which can then be subtracted more easily), than to minimize the dark current at the expense of temperature stability.[RC2] The 'darkness' required for this measurement is achieved by disconnecting the optical fibres from the spectrometer module. The dark exposures are also used to identify hot pixels, defined as pixels which return a fully saturated response independently of the amount of light incident on them. We flag all pixels with mean measured responses above a threshold value as hot pixels, and apply a threshold standard deviation to filter out pixels which are 'hot' intermittently, as the pixel response to dark input should not vary significantly over time.

We identify dead pixels, pixels which show no response when light is incident on them, using exposures where a white light source is observed. For this purpose we use the integrating sphere described earlier with a quartz tungsten halogen lamp as the observed target. All pixels with measured mean responses below a threshold value are flagged as dead pixels. We also use the same measurements to produce a map of the image of each spectral band on the detector array, that is then used to determine the pixels included when processing the measured images into radiance spectra (see Sect. 3.1). The pixels forming the band map are identified on a column-by-column basis. In each column, the twenty pixels with the highest value when illuminated (excluding hot and dead pixels) are considered to be 'in-band' for the purposes of processing the images into spectra. By taking the mean of the twenty in-band pixels in each column during the processing stage (as described in Sect. 3.1) we are able to obtain a better signal to noise ratio for the final calibrated spectra.

[Figure]

**Figure 12.** Measurements of emission lamp spectra for each GHOST band, shown as digital detector response in counts as a function of horizontal pixel number. The colours correspond to the different emission lamps listed in Table 3.

**3.3 Spectral calibration**

[revised manuscript text omitted]

Here, we only present results from a fit to the $CO_2$ and $CH_4$ spectral bands in Band 2A (coloured green and blue respectively in Fig. 19) to infer the ratio of $CO_2$ to $CH_4$, according to the proxy retrieval approach (Parker et al., 2011; Frankenberg et al., 2006)[RC1]. The advantage of this approach is the reduced sensitivity to aerosols due to a cancellation of the aerosol effects in the retrieved $CO_2$ and $CH_4$ columns, so subsequently aerosols do not have to be considered for the retrieval. In using the proxy method, we assume that very similar distributions of light paths contribute to the observed spectra at the absorption wavelengths of both the gas of interest and its proxy for the total air column, introducing the requirement that we use spectrally neighboring fitting windows. In addition, the

**Table 8.** Description of the state vector.

| Quantity | Number of elements | Description |
| --- | --- | --- |
| CH$_4$ | 1 | |
| CO$_2$ | 1 | Scaling factor (no units) applied to a priori vertical profile |
| H$_2$O | 1 | |
| HDO | 1 | |
| Temperature | 1 | Shift (degrees Kelvin) added to a priori vertical profile |
| Albedo | 5 | Coefficients of fourth order polynomial describing surface albedo (no units) as function of pixel number |
| Dispersion | 5 | Coefficients of fourth order polynomial describing wavelength (metres) as function of pixel number |
| Zero level | 1 | Wavelength independent scaling factor (no units), applied to radiance spectrum and then added to approximate effect of straylight in the instrument |
| EOFs | 3 | Wavelength independent scaling factors (no units) applied to empirical orthogonal functions (see text in Sect. 5.1) which are then added to the spectrum to account for spectral systematic errors |

method assumes that our measurement exhibits equivalent sensitivity to both gases at the heights where the greatest deviations from the expected light path occur as a result of scattering (Frankenberg et al., 2006).[RC1] The retrieved state vector elements are listed in Table 8. In future, we plan to also
530    adopt a full-physics retrieval approach where aerosol parameters are retrieved together with CO$_2$ by a simultaneous fit to the O$_2$ band (Band 1), the weak CO$_2$ (or CH$_4$) band (Band 2) and the strong CO$_2$ band (Band 3).

[revised manuscript text omitted]

580 The divergence of the $CO_2$ scaling factor away from the $CH_4$ scaling factor also results in a negative bias compared with the OCO-2 data and the CarbonTracker model output, which varies from 2 ppm up to around 8 ppm along the flight leg. Given that OCO-2 makes observations at a higher spatial density than GHOST (because of its eight pixel cross-track imaging capability), we would expect OCO-2 data to be more representative of the variability in $XCO_2$ along the overpass compared with

585 GHOST. We can conclude from this that the change in discrepancy between the two datasets along the flight track is unlikely to be a result of spatial changes in $XCO_2$ that OCO-2 could be missing as a result of mismatches in spatial resolution and location.[RC1]

When using the proxy retrieval method, we are assuming that the two spectral ranges used are close enough in wavelength space that any systematic errors are eliminated when taking the ratio.

590 However, this would not work (and could subsequently introduce the observed divergence in scaling factors obtained from the two spectral ranges used) in the presence of an instrumental error or effect which has sufficient wavelength dependence within the wavelength ranges observed by GHOST. Examples of these include unaccounted for straylight incident on that part of the detector, or a wavelength dependent error in the radiometric calibration.[RC1] Whilst we are still investigating

595 the underlying cause of this bias, we anticipate that an improved understanding of straylight effects in the instrument through ongoing laboratory measurements will help us to correct for the observed bias, as will comparison of retrieval results from future GHOST flights against those from ground-based validation sites.

**6 Summary**

600 GHOST is a new shortwave infrared grating spectrometer, which employs a unique compact design to optimize the science benefits provided (with regards to wavelength ranges covered and spectral resolution) given the small size of its optical payload. As described in Sect. 2, this has been achieved by adapting advances in optical technology originally designed for astronomy applications, namely the use of fibre bundles to transfer light from the fore-optics into the spectrometer, and the imaging

605 of multiple spectral bands onto a single detector array with a single diffraction grating. Through the laboratory testing and calibration discussed in Sect. 3, we have demonstrated that the GHOST radiometric and spectral performance is suitable for retrieving total column concentrations of greenhouse gases from the measured shortwave infrared spectra.

The CAST-ATTREX Global Hawk flight campaign in January-March 2015 (Section 4) was our

610 first opportunity to operate GHOST on board an aircraft. Following extensive engineering and communications testing at NASA Armstrong, GHOST was successfully integrated into the Global Hawk aircraft, and flew for the first time on February 26[th] 2015. During the campaign, GHOST took part in three flights on board the Global Hawk for a total duration of approximately 39 hours, maintaining vacuum and thermal stability throughout up to altitudes of 20 km above sea level. Data from

the March 10[th] 2015 flight are analyzed in Sect. 5, which shows for the first time greenhouse gas retrievals from spectra measured in-flight by GHOST.

This paper has introduced the GHOST instrument, and shown its potential for airborne greenhouse gas remote sensing through initial results from its first science flights. Our future plans for this dataset include applying a full physics retrieval to the measured spectra (as discussed in Sect. 5.1), which will allow for a more like-for-like comparison with the OCO-2 measurements during the 10[th] March 2015 overpass. In addition to the CAST-ATTREX Global Hawk flights we have also flown GHOST over the UK on board the NERC Airborne Research Facility (ARF) Dornier 228 aircraft in April and May 2015, targeting emissions hot spots. Future plans include analysis of this dataset, as well as further flights supported by the ARF in Spring 2018. These flights will investigate how multi-angle views of a scene can improve the retrieval accuracy through better representation of scattering effects in the atmosphere, as well as targeting ground-based remote sensing instruments in the UK which can be used for validation of the GHOST results. Future flights (including those planned for 2018) will benefit from a new depolariser unit which has now been installed in the TAM, eliminating the spectral oscillations that affected previous measurements (see Appendix A). In addition we are working on further calibration measurements prior to these flights, which will include characterization of straylight effects and will enhance our understanding of the instrument (for example, explaining the different spectral radiances observed in Bands 2A and 2B as seen in Fig. 19)[RC2], 
[revised manuscript text omitted]

840   Palmer, P. I., O'Doherty, S., Allen, G., Bower, K., Bösch, H., Chipperfield, M. P., Connors, S., Dhomse, S., Feng, L., Finch, D. P., Gallagher, M. W., Gloor, E., Gonzi, S., Harris, N. R. P., Helfter, C., Humpage, N., Kerridge, B., Knappett, D., Jones, R. L., Le Breton, M., Lunt, M. F., Manning, A. J., Matthiesen, S., Muller, J. B. A., Mullinger, N., Nemiitz, E., O'Shea, S., Parker, R. J., Percival, C. J., Pitt, J., Riddick, S. N., Rigby, M., Sembhi, H., Siddans, R., Skelton, R. L., Smith, P., Sonderfeld, H., Stanley, K., Stavert, A. R., Wenger, A., White, E.,

845   Wilson, C., and Young, D.: A measurement-based verification framework for UK greenhouse gas emissions: an overview of the Greenhouse gAs Uk and Global Emissions (GAUGE) project, Atmospheric Chemistry and Physics Discussions, 2018, 1–52, doi:10.5194/acp-2018-135, https://www.atmos-chem-phys-discuss.net/acp-2018-135/, 2018.

Parker, R., Boesch, H., Cogan, A., Fraser, A., Feng, L., Palmer, P. I., Messerschmidt, J., Deutscher, N., Grif-

850   fith, D. W. T., Notholt, J., Wennberg, P. O., and Wunch, D.: Methane observations from the Greenhouse Gases Observing SATellite: Comparison to ground-based TCCON data and model calculations, Geophysical Research Letters, 38, doi:10.1029/2011GL047871, http://dx.doi.org/10.1029/2011GL047871, 115807, 2011.

Peters, W., Jacobson, A. R., Sweeney, C., Andrews, A. E., Conway, T. J., Masarie, K., Miller, J. B., Bruhwiler, L. M. P., Pétron, G., Hirsch, A. I., Worthy, D. E. J., van der Werf, G. R., Randerson, J. T.,

855   Wennberg, P. O., Krol, M. C., and Tans, P. P.: An atmospheric perspective on North American carbon dioxide exchange: CarbonTracker, Proceedings of the National Academy of Sciences, 104, 18 925–18 930, doi:10.1073/pnas.0708986104, http://www.pnas.org/content/104/48/18925.abstract, 2007.

Pitt, J. R., Le Breton, M., Allen, G., Percival, C. J., Gallagher, M. W., Bauguitte, S. J.-B., O'Shea, S. J., Muller, J. B. A., Zahniser, M. S., Pyle, J., and Palmer, P. I.: The development and evaluation of airborne in situ $N_2O$

860   and $CH_4$ sampling using a quantum cascade laser absorption
spectrometer (QCLAS), Atmospheric Measurement Techniques, 9, 63–77, doi:10.5194/amt-9-63-2016, https://www.atmos-meas-tech.net/9/63/2016/, 2016.

Rodgers, C.: Inverse Methods for Atmospheric Sounding: Theory and Practice, World Sci., 2000.

Spurr, R.: LIDORT and VLIDORT: Linearized pseudo-spherical scalar and vector discrete ordinate radia-

865   tive transfer models for use in remote sensing retrieval problems, pp. 229–275, Springer Berlin Heidelberg, Berlin, Heidelberg, doi:10.1007/978-3-540-48546-9_7, https://doi.org/10.1007/978-3-540-48546-9_7, 2008.

Stopford, C., Kaye, P., Ulanowski, J., Hirst, E., Greenaway, R., Dorsey, J., Gallagher, M., Woods, S., Lawson, P., Thornberry, T., Rollins, D., and Harris, N.: AIITS: Preliminary light scattering data from Tropical Tropopause Layer cirrus, in: Composition and Transport in the Tropical Troposphere and Lower Stratosphere Meeting - Boulder, Colorado, United States, 2015.

Thompson, D. R., Leifer, I., Bovensmann, H., Eastwood, M., Fladeland, M., Frankenberg, C., Gerilowski, K., Green, R. O., Kratwurst, S., Krings, T., Luna, B., and Thorpe, A. K.: Real-time remote detection and measurement for airborne imaging spectroscopy: a case study with methane, Atmospheric Measurement Techniques, 8, 4383–4397, doi:10.5194/amt-8-4383-2015, https://www.atmos-meas-tech.net/8/4383/2015/, 2015.

Viatte, C., Lauvaux, T., Hedelius, J. K., Parker, H., Chen, J., Jones, T., Franklin, J. E., Deng, A. J., Gaudet, B., Verhulst, K., Duren, R., Wunch, D., Roehl, C., Dubey, M. K., Wofsy, S., and Wennberg, P. O.: Methane emissions from dairies in the Los Angeles Basin, Atmospheric Chemistry and Physics, 17, 7509–7528, doi:10.5194/acp-17-7509-2017, https://www.atmos-chem-phys.net/17/7509/2017/, 2017.

Wunch, D., Toon, G. C., Blavier, J.-F. L., Washenfelder, R. A., Notholt, J., Connor, B. J., Griffith, D. W. T., Sherlock, V., and Wennberg, P. O.: The Total Carbon Column Observing Network, Philosophical Transactions of the Royal Society of London A: Mathematical, Physical and Engineering Sciences, 369, 2087–2112, doi:10.1098/rsta.2010.0240, http://rsta.royalsocietypublishing.org/content/369/1943/2087, 2011.